# Light-induced quantum tunnelling current in graphene

Mohamed Sennary[1], Jalil Shah[1], Mingrui Yuan[1,2], Ahmed Mahjoub[3], Vladimir Pervak [4], Nikolay V. Golubev[1] & Mohammed Th. Hassan [1,2] ✉

In the last decade, advancements in attosecond spectroscopy have allowed researchers to study and manipulate electron dynamics in condensed matter via ultrafast light fields, offering the possibility to realise ultrafast optoelectronic devices. Here, we report the generation of light-induced quantum tunnelling currents in graphene phototransistors by ultrafast laser pulses in an ambient environment. This tunnelling effect provides access to an instantaneous field-driven current, demonstrating a current switching effect (ON and OFF) on a ~630 attosecond scale (~1.6 petahertz speed). We show the tunability of the tunnelling current and enhancement of the graphene phototransistor conductivity by controlling the density of the photoexcited charge carriers at different pump laser powers. We exploited this capability to demonstrate various logic gates. The reported approach under ambient conditions is suitable for the development of petahertz optical transistors, lightwave electronics, and optical quantum computers.

The development of ultrafast light tools is vital for studying light-matter interactions and related electron motion dynamics in real time[1–3]. For instance, the generation of XUV attosecond pulses via high-harmonic generation in the solid-state[4–7] permitted probing the strong field-induced electron dynamics in condensed matter[8–14]. Recently, the generation of a single attosecond electron pulse and the development of attomicroscopy have given access to the bound electron dynamics in the nanostructure and connected it to its morphology[15]. Moreover, the ability to manipulate and synthesis the waveform of ultrashort laser pulses allows for controlling the electronic motion, electronic structure, and physical properties of dielectric and semiconductor materials to demonstrate ultrafast optical switches[16–27]. Furthermore, both optical and XUV pulses have been used to generate ultrafast current signals[27–33]. These studies have found many applications, such as the demonstration of optical-based devices for sampling the ultrafast waveforms of light[33–52].

Recently, the generation of light-induced current ($I_L$), based on photoexcitation of graphene's carriers, has been reported[52–55]. This average $I_L$ is measured and demonstrated based on the flow of the excited carriers between two metal electrodes in a circuit. The $I_L$ has

been controlled by manipulating the carrier dynamics by changing the intensity and the carrier-envelope phase (CEP) of the pump laser pulse[53,54]. It is noteworthy to mention that the $I_L$ current has a contribution from two currents: (i) The ultrafast instantaneous *field-induced current ($I_E$)*, which is generated from the motion of the excited virtual carriers−driven by the light field−in the conduction band (intraband current) of graphene. This $I_E$ current is a transient current and lasts only during the laser pulse field time window. (ii) The *photo-induced current ($I_P$)*, which is generated due to the excitation of real carriers from the valance band to the conduction band by absorbing photon(s) from the pump pulse (interband current). Then, these excited carriers relax back to the valance band on a time scale of a few ten picoseconds. Hence, the major contribution in $I_L$ is coming from $I_P$, while the contribution of $I_E$ is minor since the latter exists in a finite time (the duration of the laser pulse in femtosecond time scale) compared to the long-time response of the current detector (in few milliseconds time scale).

In previous studies using symmetric graphene[52–55], the $I_E$ was averaging out, and the measured current was mainly from the $I_P$ contribution. Additionally, the demonstrated control by changing the CEP

[1]Department of Physics, University of Arizona, Tucson, AZ, USA. [2]James C. Wyant College of Optical Sciences, University of Arizona, Tucson, AZ, USA. [3]Jet Propulsion Laboratory, California Institute of Technology, Pasadena, CA, USA. [4]Ludwig-Maximilians-Universität München, Am Coulombwall 1, Garching, Germany. ✉e-mail: mohammedhassan@arizona.edu

is based on the modulation of the excited real carrier's density and the displacement of virtual carriers in real space by changing the pump pulse intensity[53]. Nevertheless, the detection and distinguishing of the $I_E$ haven't been measured or demonstrated yet.

In this work, we utilised a graphene-silicon-graphene (Gr-Si-Gr) phototransistor to generate sub-microamperes light-induced current ($I_L$) by few-cycle laser pulses. In our transistor, the current flows based on quantum tunnelling between the graphene and the silicon junction. Hence, the generated current is gated in time, which allows us to access and record the ultrafast instantaneous field-induced current ($I_E$). The $I_E$ modulates periodically in real-time, following the waveform of the driver field, enabling a current switching between two states (ON and OFF) with a time speed of 630 attoseconds (1.6 petahertz). Moreover, we control the $I_L$ current amplitude by increasing the induction laser beam intensity and determine the consequent enhancement of our phototransistor photoconductivity. Finally, the flexibility of our transistor setup allowed us to combine a DC current ($I_V$), generated by applying external voltage, with the $I_L$, to demonstrate several logic gates within our phototransistor. Importantly, the presented experiments are performed under ambient standard temperature and pressure conditions, making this phototransistor at the technology readiness level for developing attosecond and lightwave quantum optoelectronics.

## Results and discussion
### Light-induced quantum tunnelling current
The development of graphene field-effect phototransistor based on quantum tunnelling is essential to access the field-induced current in graphene. Hence, we optically dopped a graphene-based channel transistor to prepare a Gr-Si-Gr channel (the preparation and the operation mechanism of our device are explained in SI and illustrated in Supplementary Fig. 1). Optical microscope images of this channel and an illustration of its band structure are displayed in Fig. 1a. The Gr-Si-Gr composition is confirmed by Raman spectroscopy characterisation measurements and results (as explained in SI and shown in Supplementary Fig. 2). Initially, we maintained the external voltage ($V_{ext}$) in our device at zero voltage, ensuring that no external DC current is

generated ($I_V = 0$). Then, we focused ultrafast laser pulses (the measured temporal profile (FWHM~6.5 fs) is shown in Supplementary Fig. 3) by a parabolic mirror into the Gr-Si-Gr channel (see details in Methods). Hence, a light-induced current signal ($I_L$) —in the few hundred nanoamperes level—is generated and measured (see Fig. 1b). Note this $I_L$ signal switches OFF when the Laser beam is blocked, as shown in see Fig. 1b. When the laser is ON, the graphene charge carriers are excited, leading to an increase in their concentration. Hence, the density of states changes, causing the Fermi energy level to shift from the neutral level and increasing the voltage difference, as demonstrated elsewhere[56] (see illustration on the right side of Fig. 1a). Accordingly, the generated $I_L$ flows in our device by quantum tunnelling of the carriers from the graphene to silicon. To prove this current tunnelling, we measured the I-V curves in both cases (laser ON and OFF), as shown in inset of Fig. 1b. From these measurements, we obtained the I-V curve shown in Fig. 1c by subtracting the I-V curve when the laser is OFF from the I-V curve when the laser is ON (after shifting it to compensate for the $I_L$ offset). Remarkably, this curve (Fig. 1c) is a tunnelling characteristics I-V curve, validating our interpretation of the $I_L$ generation and flow mechanism[56,57]. Moreover, the asymmetric of this obtain I-V curve (around $V_{ext} = 0$ in Fig. 1c) suggests that the illumination of the laser beam of the two graphene sides is uneven (see Supplementary Fig. 1a), which cause the symmetry breaking and explains the flow of the light-induced current in our setup.

### Petahertz phototransistor
The current tunnelling flow mechanism gates the generated current signal in time and allows us to measure and distinguish the instantaneous field-induced current ($I_E$), which is generated due to the intraband dynamics in graphene. This current evolves during the laser pulse's existence time window. Hence, to measure $I_E$ in real-time, we opted to perform a cross-correlation current measurement between two current signals generated by two pump laser pulses. Accordingly, we modified our setup by splitting the input laser beam into two beams using a beamsplitter (Fig. 2a); each beam power has been set to have a similar estimated low field strength of ~0.85 V/nm. Then, we recorded

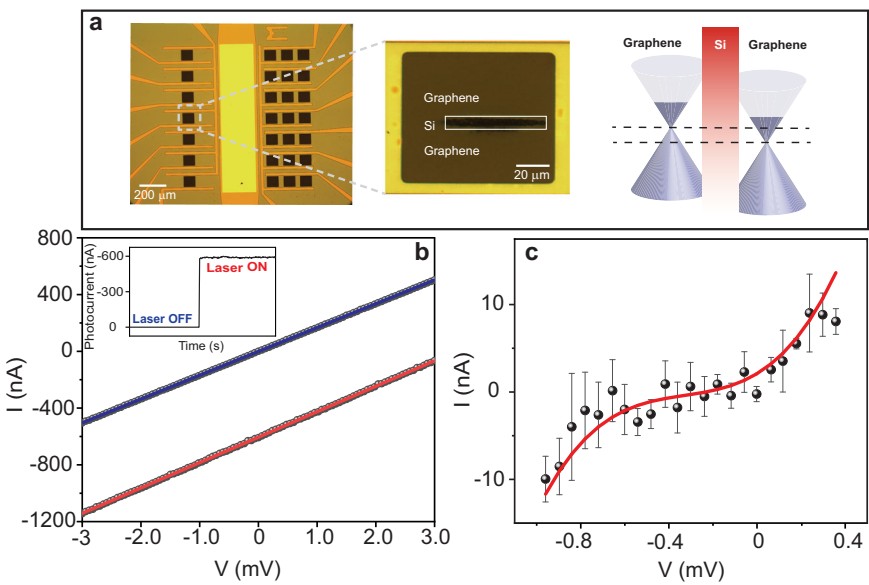

**Fig. 1 | Light-induced quantum current tunnelling in graphene phototransistor. a** The optical microscope (and zoom in) images of the graphene-silicon (Si)-graphene phototransistor and illustration of its band structure, the black dashed line presents the Fermi level. **b** The measured current-voltage (I–V) curve in

case of laser ON (blue line) and laser OFF (red line). The inset shows the switching ON and OFF the photocurrent signal by the laser beam. **c** the tunnelling characterics I–V curve for the Gr-Si-Gr transistor and the redline is an eye guide. The error bars present the calculated standard deviation error of three scans.

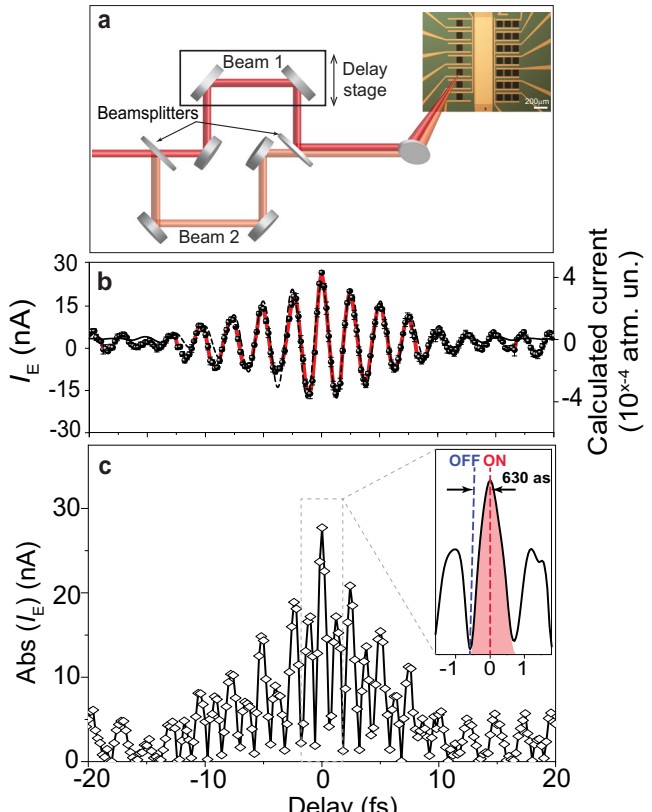

**Fig. 2 | Attosecond current switching. a** Cross-correlation current measurement setup. The pump laser beam splits into two beams by beam splitter. The two beam are focused into the transistor and generate current signals. The delay between these two generated signals is controlled by a delay stage implemented in one of the beam paths. **b** Instantaneous field-induced current ($I_E$) (average of three measurements), shown as black dots connected by red line. The error bars present the calculated standard deviation error of three scans. The calculated current is plotted in dashed black line. **c** Absolute $I_E$ measured signal modulation in time, obtained from (**b**), is plotted in diamond shape points connected with black line. The inset in (**c**) (a zoom in delay ranges from −1.5 to 1.5 fs) shows the switching of the current ON and OFF with a periodicity of 630 as.

the current as a function of the time delay between the two pulses. The recorded current, when the two pulses are not overlapping in time, is 75 nA. Our setup's capability enabled the compensation (cancelling out) of the $I_P$ current signal (generated by the interband dynamics)—which has the main contribution in $I_L$—by applying an external voltage ($V_{ext}$) value until the output measured current is zero amperes. The average of three first-order cross-correlation current measurements is shown in Fig. 2b (black dots connected with red lines). The absolute measured $I_E$ current amplitude signal in real-time (plotted in Fig. 2c) switches from 29 nA (ON status) to <1 nA (OFF status) in 630 attoseconds (see the inset of Fig. 2c), demonstrating the attosecond current switching in our phototransistor. It is noteworthy that the modulation of the $I_E$ oscillates between negative and positive values. This indicates that the $I_E$ flows alternatively from the two graphene sides to the silicon junction every half-cycle of the driver field (depending on the driver field direction) causing the switching in subfemtosecond time window.

On another note, we replaced the phototransistor by a power metre, and we observed only a 10% oscillation in the power between the two pulses at the temporal overlap, indicating a minor contribution of the current amplitude oscillation (Fig. 2b) potentially originated from the optical interference. Please note the two beams aren't colli-nearly propagating, and they incident on the sample with small angles (<5°) (Fig. 2a), which minimises the optical interference effect.

Moreover, the measurements were conducted at low power in the linear regime, where power is directly proportional to intensity. Hence, we can estimate that 10% of the current modulation (shown in Fig. 2b) may come from the optical interface and the $I_P$ change. Although, the contrast of the petahertz switching would remain in the range of 1 to 25 nA.

Furthermore, we measured the $I_E$ cross-correlation using a chirped input pulse in one beam and the 6.5 fs pulse in the other beam and the results are shown in Supplementary Fig. 4 indicating that the $I_E$ is field sensitive. In addition, $I_E$ is polarisation dependent since the signal drops when the input linear polarised beam is converted to circular polarised light (See Supplementary Fig. 5).

Hence, we attributed this measured current oscillation to the drifting and tunnelling of the excited carriers within the conduction band (intraband current) of graphene following the driver laser field[27,34,53,58,59]. To confirm our observations, we performed quantum mechanical calculations to simulate our experiment's current measurements. In our calculation, we first assumed that the measured cross-correlation current (Fig. 2b) reflects the cross-correlation of the laser fields. Hence, we decomposed the waveform of the driver pulse (plotted in the red line in Fig. 2) from the cross-correlation profile in Fig. 2a. Notably, the temporal profile of the deconvoluted waveform and the measured temporal profile of the pump pulse (Supplementary Fig. 6) are in a good agreement, validating our assumption. We utilised this waveform in our quantum simulation model after considering the tunnelling effect by adding a complex absorbing potential (CAP), as explained in Methods. Then, we calculated the generated net current after the action of the two pulses as a function of the time delay (Supplementary Fig. 7a) and plotted it (dashed black line in Fig. 2b) in contrast with the measured current. These two currents are in good agreement and follow the pump pulse waveform. Noteworthy, removing effects responsible for optical interference from our simu-lations (see Supplementary Fig. 7b and the discussion in section Methods), the generated current remains the same as Supplementary Fig. 7a. However, ignoring the tunnelling effect either by preventing the electrons to accelerate by the field (Supplementary Fig. 7c) or removing Gr–Si junction completely (Supplementary Fig. 7d), we observed that the calculated cross-correlation net current becomes zero due to the averaging out of the $I_E$ current, which confirms the pivotal role of the tunnelling in our $I_E$ measurements and explains why the previous studies were not able to access or measure this field-driven current[53,54].

## Controlling the light-induced tunnelling current in graphene

The photo-induced $I_P$ current signal has the main contribution to the $I_L$ current. $I_P$ is generated from the interband current dynamics in gra-phene. Hence, the amplitude of $I_P$ depends on the excited charge carrier's density and its distribution in the reciprocal space, which can be controlled by changing the intensity of the exciting pulse. Thus, in our experiment (setup is shown in Fig. 3a), we measured the $I_L$ amplitude at different field intensities of the pump laser ranging from 0 to 2 V/nm and plot the result in Fig. 3b (black dots connected by red line). The $I_L$ amplitude increases gradually and then reaches a plateau at a higher intensity. This can be attributed to the increase in the number of excited carriers and the population in the conduction band before the carriers reach saturation. Accordingly, we calculated the average excited carrier population at different field intensities by sol-ving the time-dependent Schrödinger equation (more details are provided in Methods) and plotted it in the blue line in Fig. 3b. The calculated carrier population exhibits dynamic behaviour and plateau similar to the measured current $I_L$ shown in Fig. 3b. Moreover, Fig. 3c shows the distribution of excited carriers pumped by -1.2 V/nm in the reciprocal space of graphene. In Fig. 3c, the ring structure around the Dirac point ($k_x = k_y = 0$) reflects the single-photon excitation region. The presence of the population in the vicinity of the Dirac point is due

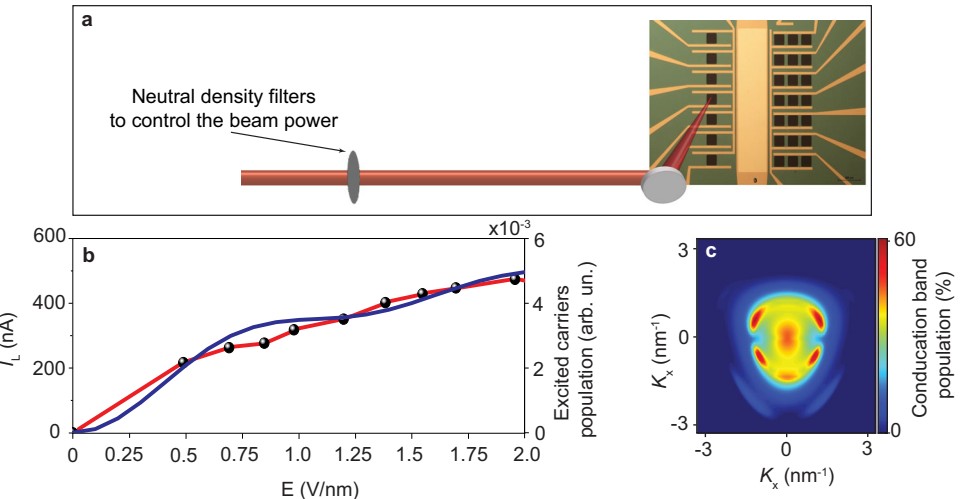

**Fig. 3 | Controlling the light-induced current signal. a** Experimental setup illustration for controlling the light induced current in a graphene-silicon-graphene transistor. The laser beam of the pump pulse is focused by a parabolic mirror into a transistor channel. The power is controlled by a neutral density filter. **b** the measured light-induced current $I_L$ (black dots connect by red line) and the calculated excited carriers' populations (blue line) as a function of the pump laser field intensity. **c** calculated carrier distribution in the reciprocal space ($K$) of graphene excited by 1.2 V/nm laser field.

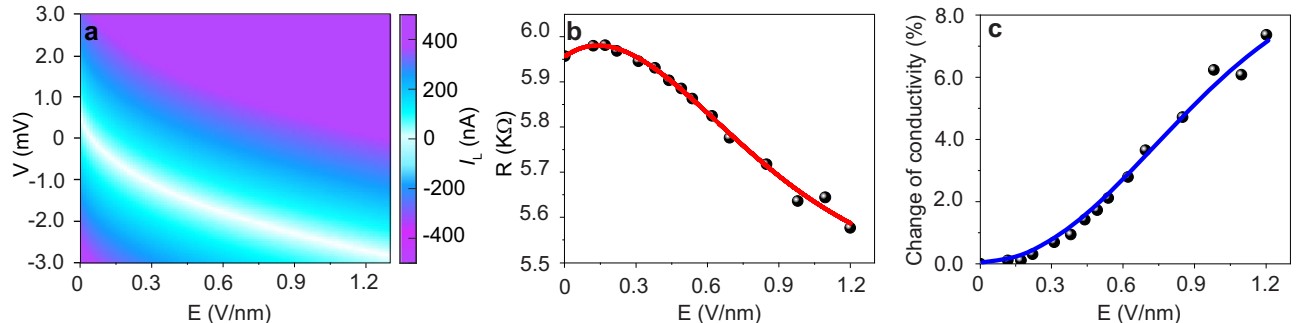

**Fig. 4 | Photoconductivity enhancement in the graphene phototransistor. a** Acquired I–V curves (average of three measurements) at different pump field intensities. **b, c** The change of resistance (R) and conductivity as function of the laser field intensities obtained from the measured I–V curves in (**a**), respectively. The red line in (**b**) is an eye guide and the blue line in (**c**) represents the calculated conductivity from our simulation model.

to the temperature effects (see Methods), which are considered in our calculations. Also, Fig. 3c shows a very minor exciting start to appear from two-photon absorption. These results explain the linear trend and the plateau in the measured current (Fig. 3b) as a saturation of the single-photon excitations. Increasing the field intensity even further is expected to show a nonlinear behaviour increase in the current due to the increase in the two-photon excitation contribution, which, however, cannot be observed in our measurement since we observed a damaging effect at higher field strength.

Next, we studied the effect of the light-induced current and carrier excitation on the resistivity and photoconductivity, mainly driven by $I_p$, of our phototransistor[60]. Thus, we measured the I–V curves at different pump laser field intensities. The results are shown in Fig. 4a. The asymmetry in the positive and negative voltage sides is due to the generation of $I_L$ with different values as the intensity increases. We focused our measurement on the intensity range before the saturation (from 0–1.2 V/nm). From the slopes of the measured I–V curves (in Fig. 4a), we calculated the resistance (R) as a function of the field intensity. The resistance of the phototransistor remains the same until a certain intensity, then it decreases from ~6 to less than 5.6 KΩ at 1.2 V/nm, as shown in Fig. 4b. Accordingly, the phototransistor conductivity increased by ~7.5% (Fig. 3c, black points). The blue line in Fig. 3c shows

the simulation fitting of the conductivity change at different intensities (as explained in the Methods).

The controlling of the $I_L$ signal (hereafter referred to as signal A) and the DC current ($I_V$) (referred to as signal B) by adjusting laser pulse intensity and the applied external voltage in our phototransistor, respectively, allow us to demonstrate various optical logic-gates. For instance, by applying $V_{ext}$ of −3.6 mV, we generated $I_V$ current of 600 nA; this effectively cancelled the induced $I_L$ current (−600 nA, shown in the inset of Fig. 1b). Consequently, our device measures no output current signal, demonstrating the XOR & NOT logic gates (see Tables 1 and 2). When adjusting the applied $V_{ext}$ such that the $I_V$ is <$I_L$, the output current signal ≠0. In this case, we can establish the logic gate OR, as shown in Table 3. Moreover, by exploiting and illuminating all seven single-graphene channels and the seven triple-graphene transistor channels in our device with different power-controlled laser beams simultaneously, we can create a multichannel phototransistor (operating with laser repetition rate) and establish all possible logic gates for developing digital quantum tunnelling-based photonics devices. Furthermore, the petahertz logic gate can be demonstrated by using $I_E$. In this case, the delay between two laser pulses $\tau$ is the input signal and the total $I_E$ is the output signal (see Fig. 2b). When the delay between the two pulses $\tau$ = 360 as, the output $I_E$ signal = 0. Alternatively,

## Table 1 | Demonstration of the XOR logic gate

| Signal A: $I_L$ | | Signal B: $I_V$ | | Signal A XOR B: Measured output current | |
|---|---|---|---|---|---|
| OFF | 0 | OFF | 0 | OFF | 0 |
| ON | 1 | ON | 1 | OFF | 0 |
| ON | 1 | OFF | 0 | ON | 1 |
| OFF | 0 | ON | 1 | ON | 1 |

Signal A is $I_L$, where ON status means the laser beam is illuminating the Gr-Si-Gr transistor and OFF status means no laser is sent to the transistor. Signal B is the applied DC current $I_V$ by the external voltage source. ON and OFF status means sending -3.6 and 0 mV signal from the source.

## Table 2 | Demonstration of the NOT logic gate

| Signal A: $I_L$ | | NOT A: Measured output current | |
|---|---|---|---|
| ON | 1 | OFF | 0 |
| OFF | 0 | ON | 1 |

We adjust the external voltage to compensate for the generated $I_L$. In this case, no $I_L$ current is generated if the laser is ON; thus, the measured output signal will be OFF. Simultaneously, the measured current will turn to be ON, when the laser signal is switched off.

## Table 3 | Demonstration of the OR logic gate

| Signal A: $I_L$ | | Signal B: $I_V$ | | Signal A OR B: Measured output current | |
|---|---|---|---|---|---|
| OFF | 0 | OFF | 0 | OFF | 0 |
| ON | 1 | ON | 1 | ON | 1 |
| ON | 1 | OFF | 0 | ON | 1 |
| OFF | 0 | ON | 1 | ON | 1 |

$I_V > I_L$, so the output current signal will be always ON unless both $I_V$ and $I_L = 0$.

## Table 4 | Demonstration of the Petahertz NOT logic gate

| Signal A: Delay $\tau$ | | Signal B: $I_E$ | |
|---|---|---|---|
| OFF | 0 | ON | 1 |
| ON | 1 | OFF | 0 |

$I_E$ is ON when the two pulse induced laser pulses are perfectly overlapped ($\tau = 0$). If the pulses are delayed by 630 as the output $I_E$ is zero.

when the delay between the two pulses $\tau = 0$ as, the output $I_E$ has 29 nA (see Fig. 2, a) which present the status 1, demonstrating the NOT gate as summarised in Table 4, open the door for establishing ultrafast optical computers.

In this work, we demonstrate the light-induced quantum tunnelling current in a Gr-Si-Gr phototransistor. The current flow is based on the tunnelling of electrons from the graphene to the Si Junction. This current has more than three orders of magnitude better efficiency than the typical graphene transistor[53,54]. Moreover, this high efficiency led to generating a decent light-induced current amplitude at low pumping laser power. Hence, this Gr-Si-Gr transistor can operate in ambient conditions (normal pressure and temperature conditions) in analogy to the typical graphene phototransistor, which operates in vacuum to avoid the oxidation of graphene and the degradation of the transistor when illuminated with a high-intense laser beam. Furthermore, the presented current tunnelling mechanism in the Gr-Si-Gr transistor gate the laser field-induced current signal; thus, it subsists after the pulse, which is not possible in a symmetric graphene transistor. Hence, this ultrafast current—which has a sub-femtosecond switching time—can be logged, demonstrating the petahertz current switching speed in our transistor. Furthermore, the tunnelling effect led to dynamic modification of the resistivity and conductivity of the phototransistor. We

report a reduction in the transistor photoresistivity by ~0.4 KΩ, which corresponds to an enhancement of 7.5% in the photoconductivity. Hence, this work promises to advance the scientific and technological advancements of ultrafast lightwave quantum electronics, attosecond optical switches, and ultrafast data encoding and communication[18,26]. Moreover, the ability to optically control the light-induced quantum current signal and establish different optical logic gates open the door for developing ultrafast quantum optical computers.

# Methods
## Experiment setup
In our setup, a 1 mJ few-cycle laser pulse centred at 750 nm is generated from an OPCPA-based (passively carrier-envelope phase (CEP) stabilised) laser system with a 20 kHz repetition rate. A supercontinuum laser beam that spans over 400–1000 nm is generated by focusing the laser beam in a hollow-core fibre (HCF). This supercontinuum enters a chirp mirror compressor to generate a ~ 6.5-fs laser pulse. The measured temporal profile using the FROG technique is shown in Supplementary Fig. 3. The laser beam is focused (beam diameter is ~50 μm) on one of the transistor channels by using a 25 mm parabolic mirror (Fig. 2a). The graphene chip is connected to an external voltage and current source/detector. This device is used to measure the light-induced current signal $I_L$ (see SI). To measure the field-induced current $I_E$, the output beam from the chirp mirror compressor splits into two beams by beamsplitter. One of the beams reflects off two mirrors mounted on a delay stage (piezo stage) with nanometre resolution and is combined with the second beam by another beamsplitter (Fig. 2a). Then, the two beams are sent to the same parabolic mirror and focus into the graphene chip. The $I_E$ is recorded as a function of the time delay between the two pulses. For the measurements shown in Supplementary Fig. 4, one of the pulses is chirped by propagating it through a thick piece of dispersive fused silica.

## Simulations of the excited carrier dynamics and generated currents in graphene
The light-induced population transfer dynamics in graphene can be obtained by solving the semiconductor Bloch equation[52,54,58]:

$$i\hbar\frac{\partial}{\partial t}\rho_{m,n}(\mathbf{k},t) = \left[E_m(\mathbf{k}_t) - E_n(\mathbf{k}_t)\right]\rho_{m,n}(\mathbf{k},t)$$

$$+ \mathbf{E}(t)\cdot\left\{\mathbf{D}(\mathbf{k}_t),\boldsymbol{\rho}(\mathbf{k},t)\right\}_{m,n} - i\left[\frac{1-\delta_{m,n}}{T_d} - W(\mathbf{k}_t)\delta_{m=n,C}\right]\rho_{m,n}(\mathbf{k},t) \tag{1}$$

where $\rho_{m,n}(\mathbf{k},t)$ denotes the matrix element of the density matrix $\rho(\mathbf{k},t)$, the commutator symbol "{}" is defined as $\{A,B\} = AB - BA$, $T_d$ is the interband dephasing time, and the meaning of the term $W(\mathbf{k})$ will be explained later in this section. The electronic energies of the bands $E_i(\mathbf{k})$ and the corresponding vector of matrices of the transition dipole moments $\mathbf{D}(\mathbf{k})$ are obtained for a two-band graphene model employing the tight-binding approximation. Equation (1) is derived assuming the validity of the dipole approximation and using the Houston basis in the velocity gauge with the crystal momentum frame evolving according to the Bloch acceleration theorem:

$$\mathbf{k}_t = \mathbf{k}_0 + \frac{e}{\hbar}\mathbf{A}(t), \tag{2}$$

where $e$ is the elementary charge and $\mathbf{A}(t) = -\int_{-\infty}^{t}\mathbf{E}(t')dt'$ is the vector potential of the corresponding applied electric field $\mathbf{E}(t)$.

We simulated the temporal evolution of the density matrix $\rho(\mathbf{k},t)$ in reciprocal space by numerically solving Eq. (1). We sampled the unit cell in the first Brillouin zone with a uniform $256 \times 256$ grid along the reciprocal lattice vectors. The initial electron density was generated by

employing the Fermi–Dirac distribution.

$$\rho_{n,n}(\mathbf{k}, t=0) = \frac{1}{\exp(E_n(\mathbf{k})/k_B T) + 1}, \tag{3}$$

where $k_B$ is the Boltzmann constant and the temperature $T$ is set to 298.15 K. The integration in the time domain is performed by the Runge–Kutta–Fehlberg method with adaptive time step control. We used a Gaussian waveform as follows:

$$E(t) = E_0 e^{-4\ln 2\left(\frac{t-t_0}{FWHM}\right)^2} \cos(\omega(t-t_0)) \tag{4}$$

with a photon energy $\omega$ of -1.5 eV linearly polarised along the C–C bonds of the graphene sample for modelling the applied electric field. The dephasing time $T_d$ is set to 10 fs.

The redistribution of the electron density between the valence and conduction bands of graphene under the influence of the time-dependent electric field affects the macroscopic properties of the material, such as the electrical conductivity. The latter per unit of volume can be obtained from the Kubo-Greenwood formula[61–63]:

$$\sigma_{\mu,\nu} = \frac{e^2}{i\hbar} \sum_n \int_{BZ} \frac{\partial \rho_n(\varepsilon)}{\partial \varepsilon}\bigg|_{\varepsilon = E_n} \frac{\partial_{k_\mu} E_n(\mathbf{k}) \partial_{k_\nu} E_n(\mathbf{k})}{\hbar\omega_0 + i\eta} d\mathbf{k}, \tag{5}$$

where the $\mu$ and $\nu$ indices denote the directions of the conductivity tensor $\boldsymbol{\sigma}$, $\omega_0$ is the frequency of the applied, in general AC, spatially homogeneous test current. The infinitesimal imaginary shift $\eta$ added to the frequency acts as a small inelastic scattering rate or relaxation rate, and the integration is performed over the entire Brillouin zone. The derivatives of the energy distribution function for the valence and conduction bands are obtained from the corresponding residual population distributions, $\rho_V(\mathbf{k})$ and $\rho_C(\mathbf{k})$, respectively, using the following relationship connecting the corresponding partial derivatives:

$$\frac{\partial \rho_n(\mathbf{k})}{\partial \mathbf{k}} = \frac{\partial \rho_n(\varepsilon)}{\partial \varepsilon}\bigg|_{\varepsilon = E_n} \frac{\partial E_n(\mathbf{k})}{\partial \mathbf{k}} \tag{6}$$

We computed the change in the electrical conductivity of the graphene sample as a function of the intensity of the applied electric field (see Fig. 4c). We used the DC test field in our simulations $(\omega_0 = 0)$ and assumed the electron relaxation rate $\eta$ to be 0.01.

In addition to the residual change in the conductivity resulting from the action of the laser pulse on the system, the instantaneous intraband current generated during the action of the field can be estimated as follows:

$$\mathbf{J}^{\text{intra}}(t) = \sum_n \int_{BZ} \rho_{n,n}(\mathbf{k}, t) \nabla_{\mathbf{k}} E_n(\mathbf{k}_t) d\mathbf{k} \tag{7}$$

The application of a single isolated laser pulse to the graphene system generates an electron current, which, however, vanishes after the action of the laser field on the system is over. Similarly, applying two delayed laser pulses to symmetric graphene and performing the scan over various delays will not generate any residual current in the system, as shown in Supplementary Fig. 7d.

However, in our case, the presence of the graphene–silicon junction breaks the symmetry of the system and potentially leads to electron tunnelling from the conduction band of graphene to silicon. To simulate the experimental measurements and to theoretically confirm that the field-induced current shown in Fig. 3a can exist, we added the possibility for electrons to tunnel from the conduction band of graphene to the silicon due to the created junction (see Fig. 1a and discussion in the main text). We added the complex absorbing potential (CAP) to Eq. (1) (the $W(\mathbf{k})$ term); this is a simple phenomenological way to account for electron leakage through a junction.

In our simulation, the CAP is chosen to be located to the right with respect to the Dirac point, such that its strength increases along the $k_x$ direction of the crystal momentum $\mathbf{k}$:

$$W(\mathbf{k}) = \beta \theta(k_x - K_x) A_x(t)(k_x - K_x)^2 \tag{8}$$

where $K_x = \frac{2\pi}{\sqrt{3}a}$ is the coordinate of the Dirac point along the $k_x$ direction, $\theta(k_x - K_x)$ is the Heaviside step function, $A_x(t)$ is the $k_x$-component of the vector potential $\mathbf{A}(t)$, and $\beta$, chosen to be 5.0 in our simulations, is the parameter controlling the strength of the CAP. In the presence of CAP, the electron density can leak from the conduction band of graphene when it is displaced by the vector potential in the positive $k_x$ direction, or the electron density, in principle, can be pulled to the system from the junction when the vector potential is negative. The presence of the graphene-silicon junction breaks the symmetry of the system and thus leads to the generation of a persistent current after the interaction with the applied laser field, as shown in the results plotted in Supplementary Fig. 7a.

The persistent electric current obtained via solution of Eqs. (1) and (7) with the field synthesised from the two delayed Gaussian waveforms can, in principle, be attributed to two concurrent effects: optical interference of the applied fields and electron tunnelling through Gr–Si junction. To decipher these two mechanisms from each other, we performed additional simulations, selectively switching off appropriate terms in Eq. (1) and analysing the obtained current. Therefore, in addition to two terminal cases described above, (a): full system with possibility of tunnelling and optical interference (Supplementary Fig. 7a), and (d): pure graphene with no junction (Supplementary Fig. 7d), we created two intermediate scenarios where we (b) prevent photon absorption by neglecting $E(t)\{\mathbf{D}(\mathbf{k}_t), \rho(\mathbf{k}, t)\}_{m,n}$ term. This way we exclude any optical interference effects since the obtained density matrix will not, by definition, contain any contributions coming from the interaction with an optical field (Supplementary Fig. 7b). Alternatively, we (c) prevent field acceleration of electrons by prohibiting changes of the crystal momentum frame in time: $\mathbf{k}_t = \mathbf{k}_0$. This way we exclude tunnelling effects since the electrons will not be able to move through the Gr–Si junction (Supplementary Fig. 7c). As one can see, removal of the optical excitations and thus the optical interference from the consideration (Supplementary Fig. 7b) has a minor effect on the possibility to generate the electric current. This is because the number of electrons driven to the conduction band by the photon absorption is negligible in comparison to those already present there due to the thermal effects (see Eq. (3)) or originating from the Landau-Zener transitions. At the same time, excluding acceleration of electrons by the field and thus dramatically reducing the possibility of tunnelling, we see nearly complete suppression of the generated current (Supplementary Fig. 7c).

## Data availability
Relevant data supporting the key findings of this study are available within the article and the Supplementary Information file. All raw data generated during the current study are available from the corresponding authors upon request. Source data are provided with this paper.

## Code availability
The analysis codes that support the study's findings are available from the corresponding authors upon request.

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

## Acknowledgements

This project is funded in part by Gordon and Betty Moore Foundation Grant **GBMF 11476** and Air Force Office of Scientific Research under award number **FA9550-22-1-0494** to M.H. Part of this work has been conducted by A. M at the JPL, Caltech, under a contract with the National Aeronautics and Space Administration (NASA). The computational part of this research is based upon High Performance Computing (HPC) resources supported by the University of Arizona TRIF, UITS, and Research, Innovation, and Impact (RII) and maintained by the UArizona Research Technologies department.

## Author contributions

M.S. and J.S. conducted the experiments and analysed the data. M.Y. and N.G. carried out the simulations and calculations. A.M. performed the sample and junction characterisations. V.P. designed and measured the optics used in the setup. M.H. conceived, supervised, and directed the study. All authors discussed the results and their interpretations and wrote the manuscript.

## Competing interests

The authors declare no competing interests.
