## [Peer Review file · Nature Communications]

Light-induced quantum tunnelling current in graphene

Corresponding Author: Professor Mohammed Hassan

Version 0:

Reviewer comments:

Reviewer #1

(Remarks to the Author)

Ultrafast switching of the current in the graphene transistor is very interesting. The theory including an absorbing potential is also interesting. However, there are several concerns (see below) that must be addressed before publication can be recommended.

First, the origin of the current is interpreted as “quantum tunneling between two graphene sides” in the main text. On the other hand, it is written in the theory part as “tunneling from the conduction band of graphene to silicon”. These two physics are completely different, and I can only agree with the latter. How large is the Si gap between the two graphene? Scale bar(s) should be added to Fig. 1. If the gap is very large (In Ref. 55, the gap is only 5 nm thick and the tunneling picture is correct.), how can electrons tunnel over a large distance within the ultrashort time scale?

Second, Fig. S1 shows that there is a 90-nm-thick SiO₂ layer between graphene and silicon. Why can the authors make the graphene-silicon-graphene transistor rather than the graphene-SiO₂-silicon-SiO₂-graphene transistor? This point must be made clearer because it is closely related to all the results and interpretations.

Third, the authors discuss two types of currents, I_E and I_p . The main claim of this article is the observation of I_E by the use of the graphene transistor. However, if my understanding is correct, the contribution of I_p is still much stronger (Extended Data 1) and the observation of I_E requires the excitation with the two pulses even with the special transistor. Moreover, I_E is not important in the results in Figs. 3-4 and Table 1-2 (the logic gates). These points should be explained more clearly and carefully. In addition, it should be described how the background signal was removed from the raw data to obtain the results in Fig. 2.

Fourth, it is written in the theory part that “the presence of the graphene-silicon junction breaks the symmetry” but the experimental sample (Fig. 1a) holds a symmetry due to the two junctions with opposite directions. Can the theory still explain the generation of current with the two junctions?

Fifth, as far as I understand from Ref. 54, the current produced in Ref. 54 (and probably in some others as well) is not from I_p . The currents are originated from the quantum interference occurring within a cycle and induced by the classical electric field (E), not photons (p). The interference prevents I_E from averaging out to zero. Authors should check the article again and modify their definitions of I_E and I_p , if necessary. On page 5, there is a sentence on the origin of currents in this work, “drifting of the excited carriers ...following the driver laser field”. If this sentence explains the physics of I_E , the currents in Ref. 54 should be I_E , in my opinion. In addition, there is a question how is this drift picture related to the tunneling?

Below are minor comments:

- (1) The authors should compare the achieved switching time (630 as) to the other works on the petahertz/light-wave electronics. In addition, “room temperature” and “ambient pressure” are emphasized. How are the relevant experiments done before?
- (2) Regarding “cross-correlation” on page 5, is it the first-order cross-correlation or the second-order?
- (3) Did the authors confirm the laser-polarization-angle dependence? There must be no I_E current when the polarization is parallel to the Si stripe.
- (4) Authors should add an explanation as to why the Raman spectrum in the supplement shows that there is no graphene

oxide.

(5) Is the result in the inset of Fig. 2b measured or calculated? If this is calculated, it should be clearly stated that the number was obtained by calculation.

(6) It should be stated that the theory considers only two electronic bands.

(7) Label should be added to the horizontal axis of Fig. 1b.

Reviewer #2

(Remarks to the Author)

The manuscript presents very interesting experimental results on the measurement of ultrafast current across a Gr-Si-Gr junction upon photoexcitation. When the few cycle laser pulses are incident the junction reports a several hundred nano ampere level of current. There is an oscillation on the measured current that corresponds to the quarter of the period of the driving laser field. The maximum current is defined as ON and the minimum current is defined as OFF. So, there is a 630 as of delay between the ON peak and OFF dip, as shown in figure 2. They also present IV curve in figure 3, where the current shows an increase with respect to the applied higher peak intensity of the laser in the form of its field strength.

Through a cross-correlation method they claim to also separate the instantaneous field induced current (IE) from the photo-induced current (IP) although I find the methodology is not clearly presented.

The IV curve is described as the device performing like a transistor although unfortunately I do not find any specific features of the IV curves of a transistor such as an increase followed by a saturation of the current after certain amount of voltage. Also, the cross-correlation scheme could be clearer if they sketch some pictures of the setup showing two pulses.

If the measured current is the tunneling current across the interface the calculations should also try to calculate the tunneling across Si-Ge interface. It seems to me that the calculated current densities are intra-band current densities on graphene. Presented calculation results do not seem to support the main claim of the manuscript although the intra-band current density maps on graphene look pretty.

If the current is a field-driven current, I would expect it has strong directionality with respect to the polarization direction of the laser. Have they observed anything like that?

Also, the photon energy is above the band-gap of silicon so I am not sure how the photo-excited carriers are not dominating. If the photon-energy of the light is smaller than the indirect band-gap of silicon then it would be easier to understand.

In overall, in the present manuscript, while I find interesting experimental observations, I feel that the manuscript lack a clear interpretation of the data, and I do not find any similarities with IV curves of a typical transistor.

Reviewer #3

(Remarks to the Author)

Key results

A) The photoresponse of the planar-graphene tunneling phototransistor designed in this study is a result from the tunneling of photoexcited carriers between the graphene layers of a Gr-Si-Gr transistor

B) Two components contributing to the total photoresponse are the direct tunneling of photoexcited charge carriers and the field-induced tunneling current which is carrier motion driven directly by the light field itself. The latter is attosecond scale response.

C) There is a notable photoconductivity enhancement at higher light field strength

D) This phototransistor is at the technology readiness level for developing attosecond and lightwave quantum optoelectronics

Comments on A)

Based on the current profile in Fig 1C and also the geometry of the device being graphene separated by a gap, it is reasonable to believe the current is due to tunneling photoexcited carriers. This device response is also consistent with other non-photoexcited lateral graphene tunneling devices (J. H. Yang et al., "Geometrically Enhanced Graphene Tunneling Diode with Lateral Nano-Scale Gap," IEEE Electron Device Letters, vol. 40, no. 11, pp. 1840-1843, Nov. 2019, doi: 10.1109/LED.2019.2940818.). However, clarification on device configuration would be helpful.

It would be helpful to the reader if the schematic in Fig S1a reflects the actual geometry of the device with the tunneling junction in the middle along with its dimensions of this junction. The classification of this being a graphene-silicon-graphene photoresistor is also unclear, since the methods described in the text for creating the devices involve removing the middle section of the graphene and SiO₂ which means the Si is not in contact with the graphene in the lateral direction. Based on this description it would make the device a Graphene-vacuum/air-Graphene device. This may or may not affect the simulation outcomes.

It would also be helpful to the reader to have the focused laser spot size and location described along with the tunneling junction dimensions. The size of the laser spot and also its location is important in understanding the origin of the tunneling current. For example, if the laser is focused such that it is only on one side of the graphene junction, then one might expect to only have field-induced current in one direction, so either it is always positive or always negative. This would help with visualizing the behavior of the field induced current.

Comments on B)

The tunneling configuration describe in this manuscript is expected to capture the field-induced current generated in graphene, since the junction would capture any charges moving into the junction and not away from it thus breaking the symmetry of the field-induced motion of the carriers. Moreover, this field-induced current has been observed previously in graphene under similar field intensities as such one would expect a field-induced current to contribute to the total photoresponse in the configuration as well. A petahertz responsive device with a simple easily reproducible geometry is indeed instrumental in studying sub-femtosecond dynamics.

On the other hand, more clarification maybe needed when describing the origins of the oscillating behavior of the field-induced current. The method in which the field-induced current was extracted was by removing the photo-induced current. This was done so by applying an external voltage across the contact of the device such that the total current is zero when the laser pulses are not overlapping. However, the magnitude of the photo-induce current could potentially vary when the laser pulses overlap due to the interference of the pulses which mean the photo-induced current may not be completely removed by applying a fixed voltage across the device. This would lead to a photo-induced current which also varies in time-delay of the pulses as the average number of excited carriers changes when the pulse interfere constructively or destructively. Thus, this current would also modulate similarly to the cross-correlation profile of the pulses. An explanation of the potential changes in the photo-induce current, or lack thereof, would be helpful to further solidify the conclusions of the field induced current.

Comments on C)

Increase of the photoconductivity with intensity is expected due to excitation of carriers into the conduction band, this is also confirmed with the model calculations which seems reasonable. This gives a good metric for determining the efficiency of the logic circuit since it is expected to operate at high field intensities.

Comments on D)

The technological readiness of this device would be clearer when the comments for the extraction of the field-induced current is addressed. The fabrication of the device also appears to be able to be scaled up given that a high-power scanning laser could produce the tunneling junction required. It would be helpful to give examples of methods to distinguish the ultrafast-intra-band current response from the slower overall inter-band current. As in how would one suppress the IE signal which changes sign to achieve an ultrafast logic gate states as described in table 1 and 2.

Version 1:

Reviewer comments:

Reviewer #1

(Remarks to the Author)

The authors adequately replied to many of my comments. I now understand physics better. The directional current in Fig. 1b is produced due to the unequal laser intensities on the two graphene-silicon interfaces. The offset of 75 nA in Fig. 2b is canceled by an external DC voltage, and therefore the direction of the measured current (29 nA amplitude) could be reversed. On the other hand, there are two remaining points which should be made clearer before publication.

1. The structure of the sample is still not clear to me. The authors removed graphene and SiO₂ and exposed the silicon substrate. How can it make a graphene-silicon interface? From a three-dimensional perspective, isn't there at least 90 nm distance (=SiO₂ thickness) between graphene and silicon?

2. Regarding the 'first-order' cross-correlation, I mentioned 'cross-correlation' at a different place, which is in the sentence 'In our calculation, ... cross-correlation of the laser fields'. The question was asked because it was a bit of a surprise to me that the observed result in Fig. 2b can well be approximated by the first-order cross-correlation of the laser fields, meaning that the switching speed of the device is solely determined by the cycle period time of the laser field.

The authors excluded the contribution of optical interference (reply to reviewer #3) but the revised manuscript states that 7.5 nA (= 10% of the 75 nA) of the 29 nA can come from the optical interference. On this point, the authors need to prove more carefully that the dominant effect is still the tunneling, not the optical interference. My specific comments are (2-a) An explanation should be added as to how and where the 10% power modulation was measured. Since two non-collinear beams can produce delay-dependent interference fringes in space, the local intensities on the graphene-silicon interfaces can be modulated by more than 10%. Since the current is driven by unequal local intensities, the interference fringes may cause a strong influence. Did the authors exclude this possibility, too?

(2-b) Is the noncollinear geometry considered in the theory with the dipole approximation? If not, the theory gives a strong optical interference effect and cannot support the interpretation. Additional explanation is needed on this point.

Reviewer #3

[Editorial Note: The attachment is displayed at the end of the file]

(Remarks to the Author)

The follow up comments below should be addressed before publication can be recommended.

The main claims of the manuscript is restated below for reference, follow up comments follows.

- A) The photoresponse of the planar-graphene tunneling phototransistor designed in this study is a result from the tunneling of photoexcited carriers between the graphene layers of a Gr-Si-Gr transistor.
- B) Two components contributing to the total photoresponse are the direct tunneling of photoexcited charge carriers and the field-induced tunneling current which is carrier motion driven directly by the light field itself. The latter is attosecond scale response.
- C) There is a notable photoconductivity enhancement at higher light field strength
- D) This phototransistor is at the technology readiness level for developing attosecond and lightwave quantum optoelectronics.

Follow up Comments on A)

The added scale for device in the revised manuscript is helpful. However the overall geometry of the device is still unclear. I am unsure as to why the Graphene would be incontact with the silicon since the graphene should be ontop of the SiO₂ from the fabrication process. A cross section schematic of the strucutre would be helpful here (see attached example). As for the Raman spectroscopy, while the silicon and graphene peaks are shown, that could potentially mean both the graphene and silicon are within the beamsport of the laser, but they could be vertically displaced.

Follow up Comments on B)

Since the light is exciting both graphene and silicon, is there a difference between the carriers tunneling from silicon to graphene vs graphene tunneling to silicon?

Also, in refernce to the polarization testing mentioned by reviewer #1. It is not immediately clear why the circular polarization does not produce the same variation with since you can consider it as two orthogonal linear polarizations (one parallel to the stripe, and another perpendicular to the stripe)

Lastly the linear polarization dependence is a strong experimental evidence for this affect being field driven and should be included in the main text.

Version 2:

Reviewer comments:

Reviewer #1

(Remarks to the Author)

The authors have adequately responded to my last comments. But the new data, Extended Data 1, is difficult to understand. If my understanding is correct, the two pulses are nearly identical. If so, the same signals must be observed at plus and minus delay times because the two pulses are indistinguishable regardless of the chirp and the nonlinearity involved. If the two pulses are significantly unequal, the new result would not strongly support the claim. Additional explanation(s) should be added.

Reviewer #3

(Remarks to the Author)

The comments on two of the main claims in the manuscripts was addressed in the previous rounds of comments. However more clarification is needed before publication to support the following two claims listed below (restated from the first round of comments for reference).

- A) The photoresponse of the planar-graphene tunneling phototransistor designed in this study is a result from the tunneling of photoexcited carriers between the graphene layers of a Gr-Si-Gr transistor.
- B) Two components contributing to the total photoresponse are the direct tunneling of photoexcited charge carriers and the field-induced tunneling current which is carrier motion driven directly by the light field itself. The latter is attosecond scale response.

(Follow up comments on A)

Both reviewer #1 and myself asked for clarification of the structure, which the authors has provided. While the provided explanation could explain the formation of the structure it is only a speculation on the melting of the Si. Perhaps an AFM measurement of the surface can show this effect.

(Follow up comments on B)

A clarification on the asymmetric time delayed data using a chirped pulse, and a demonstration of the time-delay photocurrent data with different polarization is needed to support this effect being field driven.

A more clear explanation on why an asymmetric current modulation implies a field driven effect is needed. This asymmetry suggests the charge carriers care about which of the two identical pulses arrives first, which intuitively even with a field driven affect should not occur.

Data showing the diminished tunneling current with respect to excitation by a pulse with linear-polarization parallel or semi parallel to the channel as mentioned in a previous rounds of comments should be included in the manuscript for this would be one of the hallmarks of this effect being field driven.

Version 3:

Reviewer comments:

Reviewer #1

(Remarks to the Author)

Regarding the new information that a combination of chirped and un-chirped pulses was used for Extended Data 1, I cannot follow why and how the asymmetry in the data is connected to their conclusion of "minor effect from the optical interference". Optical interference can give asymmetric data because one of the pulses is asymmetric in time. A clear explanation should be added before publication.

In addition, the authors should describe in the method section how the pulse was chirped.

If the asymmetry in data cannot exclude the possibility of optical interference, the measurement result with a power meter (10% modulation) is the only direct evidence. To strengthen the authors' claim, I suggest adding a description of how this number (10%) is translated into the amount of modulation of I_E . If the modulation of I_E is proportional to the laser intensity, I_E should then be modulated by 10% due to optical interference. However, if I_E varies nonlinearly with laser intensity, optical interference causes stronger modulation.

Lastly, although the authors replied to referee #3 and me previously that the connection between graphene and Si was confirmed by Raman measurements, I cannot find any proof that the SiO₂ layer was removed. There is no Raman peaks assigned to SiO₂ in Fig. S2. The authors should add an explanation before publication. If the removal of the SiO₂ layer is a hypothetical level, it should be expressed as such.

Reviewer #3

(Remarks to the Author)

The authors have adequately answered all my questions.

Response to reviewers' comments – NCOMMS-24-50952

The referees' comments are shown in bold and Italics, followed by the corresponding response in blue and the change in the revised version.

REVIEWER COMMENTS

Reviewer #1 (Remarks to the Author):

Ultrafast switching of the current in the graphene transistor is very interesting. The theory including an absorbing potential is also interesting. However, there are several concerns (see below) that must be addressed before publication can be recommended.

We would like to thank the reviewer for finding our work interesting. We appreciate the opportunity to address all the reviewer's concerns.

1- First, the origin of the current is interpreted as “quantum tunneling between two graphene sides” in the main text. On the other hand, it is written in the theory part as “tunneling from the conduction band of graphene to silicon”. These two physics are completely different, and I can only agree with the latter. How large is the Si gap between the two graphene? Scale bar(s) should be added to Fig. 1. If the gap is very large (In Ref. 55, the gap is only 5 nm thick and the tunneling picture is correct.), how can electrons tunnel over a large distance within the ultrashort time scale?

Thank you for the reviewer's comment! The Si gap is estimated to be 4.3 μm , and a scale bar has been added to the revised version of Fig. 1. We agree with the reviewer's suggestion that the tunneling is taking place in the interfaces between graphene and Si. In this studied system we have two Gr-Si interface. To clarify this point we, and we have revised the manuscript accordingly:

Page 3:

⇒ **“In our transistor, the current flows based on quantum tunnelling between the graphene and the silicon junction.”**

Page 4:

⇒ **“Accordingly, the generated I_L flows in our device by quantum tunnelling of the carriers from the graphene to silicon.”**

Page 5:

⇒ **“It is noteworthy that the modulation of the I_E oscillates between negative and positive values. This indicates that the I_E flows alternatively from the two graphene sides to the silicon junction every half-cycle of the driver field (depending on the driver field direction) causing the switching in subfemtosecond time window.”**

Page 8:

⇒ **“The current flow is based on the tunnelling of electrons between the graphene to the Si Junction.”**

2- *Second, Fig. S1 shows that there is a 90-nm-thick SiO₂ layer between graphene and silicon. Why can the authors make the graphene-silicon-graphene transistor rather than the graphene-SiO₂-silicon-SiO₂-graphene transistor? This point must be made clearer because it is closely related to all the results and interpretations.*

The SiO₂ has a large band gap of 9 eV, which prevents electron tunneling (our pump pulse is centered at 1.65 eV). As a result, the graphene-SiO₂-silicon-SiO₂-graphene transistor doesn't allow photo-induced current to flow and, therefore, cannot be measured by our setup. We have clarified this point in the revised supplementary information (SI), page 2, thanks to the reviewer's suggestion.

⇒ *To generate the Gr-Si-Gr channel, a high-power laser beam (300 mW) is focused into the centre of the graphene channel. Then, we created a thin layer (junction) of Si by depleting the thin layers of graphene and SiO₂. If the SiO₂ layer (see Fig. S1) —which is an isolator with high bandgap (9 eV)— is not totally depleted, the photo-induced current won't tunnel, flow, and generate a measurable current in our transistor setup.*

3- *Third, the authors discuss two types of currents, I_E and I_p. The main claim of this article is the observation of I_E by the use of the graphene transistor. However, if my understanding is correct, the contribution of I_p is still much stronger (Extended Data 1) and the observation of I_E requires the excitation with the two pulses even with the special transistor. Moreover, I_E is not important in the results in Figs. 3-4 and Table 1-2 (the logic gates). These points should be explained more clearly and carefully. In addition, it should be described how the background signal was removed from the raw data to obtain the results in Fig. 2.*

We thank the reviewer for the comment! We would like to clarify that in our experiment setup the main contribution of I_p signal was compensated (canceled out) by applying external voltage with the same value but different direction. To avoid any confusion we considered the reviewer's comment in the revised version of the manuscript in many positions. i.e. Page 5

⇒ *“Our setup's capability enabled the compensation of the I_p current (generated by the interband dynamics)—which has the main contribution in I_L—by applying an external voltage (V_{ext}) value until the output measured current is zero amperes.”*

It is also mentioned in the paragraph discussing the results in Figure 3-4, in page 6

⇒ *“The photo-induced I_p current signal has the main contribution to the I_L current.”*

in page 7

⇒ *“Next, we studied the effect of the light-induced current and carrier excitation on the resistivity and photoconductivity, mainly driven by I_p, of our phototransistor⁵⁹.”*

The explanation of the “how is the background signal was removed” is explained in page 5.

⇒ *“Accordingly, we modified our setup by splitting the input laser beam into two beams using a beamsplitter (Fig. 2a); each beam power has been set to have a similar estimated low field strength of ~0.85 V/nm. Then, we recorded the current as a function of the time delay between the two pulses. The recorded current, when the two pulses are not overlapping in*

time, is 75 nA. Our setup's capability enabled the compensation (cancelling out) of the I_p current signal (generated by the interband dynamics)—which has the main contribution in I_L —by applying an external voltage (V_{ext}) value until the output measured current is zero amperes.

Thanks to the reviewer's comment and suggestion! we show a case of using I_E in demonstrating the logic gate NOT. See page 7-8 and table 4 in the revised version of the manuscript.

⇒ *“Furthermore, the petahertz logic gate can be demonstrated by using I_E . In this case, the delay between two laser pulses τ is the input signal and the total I_E is the output signal (see Fig. 2b). When the delay between the two pulses $\tau= 360$ as, the output I_E signal=0. Alternatively, when the delay between the two pulses $\tau= 0$ as, the output I_E has 29 nA (see Fig. 2 a) which present the status 1, demonstrating the NOT gate as summarized in table 4, open the door for establishing ultrafast optical computers.”*

4- *Fourth, it is written in the theory part that “the presence of the graphene-silicon junction breaks the symmetry” but the experimental sample (Fig. 1a) holds a symmetry due to the two junctions with opposite directions. Can the theory still explain the generation of current with the two junctions?*

First, we would like to clarify the following: by "the presence of the graphene-silicon junction breaks the symmetry," we are referring to the disruption of the structural symmetry of the graphene layer caused by the junction. However, the symmetry breaking those results in the propagation of the light-induced current arises from the asymmetric illumination of the laser beam on both sides of the graphene (see the revised Fig. 1Sa).

Theoretically, we modeled the tunneling of electrons by a simple CAP placed in a graphene unit cell located near the junction. Importantly, within this formalism the tunneling depends on the strength of the applied vector potential (see Eq. (8) of the main text). Accordingly, if two junctions with identical but reversed in direction CAPs are included in the simulations, we still will see the generated current. This is because the applied vector potential with a certain local strength will push or pull certain number of electrons through the Gr-Si junction on one side and will do the same but with decreased/increased probability through another Si-Gr junction where the local strength of the vector potential is lower/higher. This is further explained with an illustration in the next point (Point 5).

Thanks to the reviewer comment! We clarify this in the revised manuscript in page 4 we added

⇒ *Moreover, the asymmetric of this obtain IV curve (around $V_{ext}=0$ in Fig. 1c) suggests that the illumination of the laser beam of the two graphene sides is uneven (see Fig. S1a), which cause the symmetry breaking and explains the flow of the light-induced current in our setup.*

5- *Fifth, as far as I understand from Ref. 54, the current produced in Ref. 54 (and probably in some others as well) is not from I_p . The currents are originated from the quantum interference occurring within a cycle and induced by the classical electric field (E), not photons (p). The interference prevents I_E from averaging out to zero. Authors should check the article again and modify their definitions of I_E and I_p , if necessary. On page 5, there is a sentence on the origin of currents in this work, “drifting of the excited carriers ...following the driver laser field”. If this sentence explains the physics of I_E , the currents in Ref. 54*

should be I_E , in my opinion. In addition, there is a question how is this drift picture related to the tunneling?

We would like to thank the reviewer for this insightful comment. As we understand it, the light-induced current results presented in Ref. 54 (Fig. 2), which illustrates how the generated current varies as a function of field strength and CEP. According to the discussion in Ref. 54, “*The CEP-dependent conduction-band population can be estimated by treating the electron dynamics in a fully coherent manner when the carrier decay timescales are longer than the optical cycle, which is satisfied here. The residual current is obtained from the conduction-band population distribution and the electron velocity given by the slopes of the bands, with the assumptions of a ballistic carrier lifetime of 40 fs and a diffusive decay length of 350 nm, consistent with previous literature^{12–14} (see Supplementary Methods).*”, Based on this, the primary change in current is attributed to the conduction-band population distribution after the laser pulse.

We completely agree with this explanation; however, we note that it does not fully confirm that the CEP dependence is exclusively related to the generated induced electric field (I_E), as the authors did not provide results from an experiment that would exclude changes in the peak intensity of the pulse when the CEP is changing. Given the fact that the pulse duration is very short (5.4 fs), a change of π in the CEP would induce an intensity value by approximately 20–30%. Therefore, the observed change in the generated current could also be related to the intensity dependence.

Importantly, we would like to mention that in a follow-up work (Ref. 53), the authors of Ref. 54 did indeed present results showing field-induced currents due to interference occurring within a cycle, which breaks the symmetry in graphene, which could provide further context for understanding the phenomenon.

Regarding the drifting, we believe that the charge carriers are first drifted in the conduction band by the driving field (vector potential), generating an intraband current. These carriers then tunnel from graphene into silicon. Depending on the field direction, the carriers tunnel from graphene 1 to silicon or from graphene 2 (depends on the field direction) to silicon (see illustration below in Fig. R1).

Additionally, the measurements in Fig. 1c (where the obtained IV curve is not symmetric around zero) and Fig. 2b (where the average $+I_E$ is greater than the average $-I_E$) suggest that the laser illumination on both graphene sides is asymmetric. This asymmetry is responsible for breaking the symmetry of the system, which, together with the tunneling process, leads to measure the induced electric (I_E) current signal.

Fig. R1. The electron tunneling from graphene to silicon based on the field direction.

Hence, we considered the reviewer’s suggestion and revised our explanation to be more accurate in page 5 in our revised manuscript

⇒ **“We attributed this measured current oscillation to the drifting and tunnelling of the excited carriers within the conduction band (intraband current) of graphene following the driver laser field”.**

Below are minor comments:

(1) The authors should compare the achieved switching time (630 as) to the other works on the petahertz/light-wave electronics. In addition, “room temperature” and “ambient pressure” are emphasized. How are the relevant experiments done before?

For example, our work can be compared to one of the most recognized in the petahertz/light-wave electronic work reported in Nature 605, 251-255, (2022). The demonstrated switching from 0 A to 4 pA is $\pi/2$ of the driver pulses which is estimated to be 1.4 fs, while in our work the switching speed is 0.630 fs and with a contrast from 0 to 29 nA. In addition, the work demonstrated in Nature 605, 251-255, (2022) is done at room temperature (similar to our presented work) but under vacuum conditions (1×10^{-8} Pa), while our work is done at atmospheric pressure, as mentioned in the manuscript conclusion paragraph (page 8):

In this work, we demonstrate the light-induced quantum tunnelling current in a Gr-Si-Gr phototransistor. The current flow is based on the tunnelling of electrons between the graphene to the Si Junction. This current has more than three orders of magnitude better efficiency than the typical graphene transistor^{53,54}. Moreover, this high efficiency led to generating a decent light-induced current amplitude at low pumping laser power. Hence, this Gr-Si-Gr transistor can operate in ambient conditions (normal pressure and temperature conditions) in analogy to the typical graphene phototransistor, which operates in vacuum to avoid the oxidation of graphene and the degradation of the transistor when illuminated with a high-intense laser beam. Furthermore, the presented current tunnelling mechanism in the Gr-Si-Gr transistor gate the laser field-induced current signal; thus, it subsists after the pulse, which is not possible in a symmetric graphene transistor. Hence, this ultrafast current—which has a sub-femtosecond switching time—can be logged, demonstrating the petahertz current switching speed in our transistor.

(2) Regarding “cross-correlation” on page 5, is it the first-order cross-correlation or the second-order?

It is a first-order cross-correlation since our work is done under a field strength drive one-photon excitation as shown in Fig. 3b (now Fig. 3c). This information is now included in the revised version of the manuscript, Page 5

“The average of three first-order cross-correlation current measurements is shown in Fig. 2a (black dots connected with red lines).”

(3) Did the authors confirm the laser-polarization-angle dependence? There must be no I_E current when the polarization is parallel to the Si stripe.

We appreciate the reviewer’s suggestion and have conducted the proposed test. Specifically, when we changed the polarization to be parallel to the Si stripe, we observed a significantly reduced I_E ; however, it did not vanish entirely. This behavior can be attributed to the 6.5-fs pump pulse used in our experiment, which is generated via the nonlinear propagation of multicycle pulses in a hollow-core fiber (HCF), as described in the Methods section. Consequently, the output beam from the HCF does not exhibit perfectly pure linear polarization (our measurements indicate a polarization ratio of 85% to 15%).

To address this, we introduced a polarizer to improve the beam’s linear polarization. However, the polarizer (10 mm thick) introduced temporal broadening of the pulse to approximately 200

femtoseconds, rendering the results inconclusive. This is because the observed signal disappearance could either result from the extended pulse duration or the change in polarization. As an alternative, we employed a thin quarter-wave plate to generate a circularly polarized beam. In this case, the current oscillation I_E signal disappeared then reappeared again when switching back to a linearly polarized beam.

(4) Authors should add an explanation as to why the Raman spectrum in the supplement shows that there is no graphene oxide.

The ionization energy of graphene is 4.5-5 eV and for oxygen is 13.6 eV so the laser intensity at the focus (50 μm) doesn't have enough intensity to ionize graphene and oxygen through multiphoton excitation to induce the oxidization reaction. Hence, the Raman spectrum show no graphene oxide. This explanation is added to the revised version of the SI.

(5) Is the result in the inset of Fig. 2b measured or calculated? If this is calculated, it should be clearly stated that the number was obtained by calculation.

The inset is a zoom-in from the measured data on Fig. 2b (now Fig. 2c in the revised version). Thanks to the reviewer! this now included in Fig. 2 caption to avoid any confusion for the reader in the revised version of the manuscript.

(6) It should be stated that the theory considers only two electronic bands.

The reviewer is considered and included in the revised version of the manuscript page 9

“The light-induced population transfer dynamics in graphene can be obtained by solving the semiconductor Bloch equation considers only two electronic bands”.

(7) Label should be added to the horizontal axis of Fig. 1b.

Thanks to the reviewer! The note is considered in the revised version of Fig. 1.

Finally, we would like to express our gratitude for the reviewer's comments and suggestions, which have greatly helped us improve both the manuscript and the discussion of our results. We hope we have adequately addressed all the raised questions, and we sincerely appreciate the reviewer's recommendation for publishing our manuscript in Nature Communications.

Reviewer #2 (Remarks to the Author):

The manuscript presents very interesting experimental results on the measurement of ultrafast current across a Gr-Si-Gr junction upon photoexcitation. When the few cycle laser pulses are incident, the junction reports a several hundred nano ampere level of current. There is an oscillation on the measured current that corresponds to the quarter of the period of the driving laser field. The maximum current is defined as ON and the minimum current is defined as OFF. So, there is a 630 as of delay between the ON peak and OFF dip, as shown in figure 2. They also present IV curve in figure 3, where the current shows an increase with respect to the applied higher peak intensity of the laser in the form of its field strength.

We thank the reviewer for finding our experimental results very interesting. We are happy to address all the points raised by the reviewer to further improve our presentation.

1- Through a cross-correlation method they claim to also separate the instantaneous field induced current (I_E) from the photo-induced current (I_P) although I find the methodology is not clearly presented.

We have revised the explanation of the related paragraph in the revised version, see page 5

⇒ *“Hence, to measure I_E in real-time, we opted to perform a cross-correlation current measurement between two current signals generated by two pump laser pulses. Accordingly, we modified our setup by splitting the input laser beam into two beams using a beamsplitter (see Methods and Fig. S.3c); each beam power has been set to have a similar estimated low field strength of ~ 0.85 V/nm. Then, we recorded the current as a function of the time delay between the two pulses. The recorded current, when the two pulses are not overlapping in time, is 75 nA. Our setup's capability enabled the compensation of the I_P current (generated by the interband dynamics)—which has the main contribution in I_L —by applying an external voltage (V_{ext}) value until the output measured current is zero amperes. The average of three first-order cross-correlation current measurements is shown in Fig. 2a (black dots connected with red lines). A minor contribution of the current amplitude oscillation (Fig. 2a) potentially originated from the optical interference, since we observed only a 10% oscillation in the power between the two pulses at the temporal overlap. Please note the two beams aren't collinearly propagating, and they incident on the sample with small angles ($< 5^\circ$) (Fig. S2c), which minimises the optical interference effect. Furthermore, the absolute measured I_E current amplitude signal in real-time (plotted in Fig. 2b) switches from 29 nA (ON status) to < 1 nA (OFF status) in 630 attoseconds (see the inset of Fig. 2b), demonstrating the attosecond current switching in our phototransistor.”*

We will happily consider any further specific suggestion from the reviewer to clearly more the cross-correlation measurements if it is needed.

2- The IV curve is described as the device performing like a transistor although unfortunately, I do not find any specific features of the IV curves of a transistor such as an increase followed by a saturation of the current after certain amount of voltage. Also, the cross-correlation scheme could be clearer if they sketch some pictures of the setup showing two pulses.

We thank the reviewer for their helpful comment! We would like to clarify that the IV curve presented in Fig. 1c is used to illustrate the tunneling effect, as previously reported in Fig. 3A of Science 335, 947-950, (2012) ("Field-Effect Tunneling Transistor Based on Vertical Graphene Heterostructures"). Our IV measurement in Fig. 1c shows that the current (I) increases linearly with voltage (V), saturates in the range of -0.6 to 0 mV, and then resumes a linear increase as the voltage rises further.

We are grateful for the reviewer's insightful suggestions, which have guided important revisions. Specifically:

- Fig. 2 has been updated by moving Fig. S3c from the Supplementary Material to become Fig. 2a, as recommended.
- The IV curve, previously included in Extended Data 1, has been relocated to Fig. 1b in the revised manuscript to provide a clearer representation of the typical behavior of a Graphene Field-Effect Transistor, as shown here.
- Fig. 3 has been revised to better illustrate the experimental setup, improving clarity in accordance with the reviewer's advice.

We believe these updates significantly enhance the presentation of our results and appreciate the reviewer's constructive feedback.

3- *If the measured current is the tunneling current across the interface the calculations should also try to calculate the tunneling across Si-Ge interface. It seems to me that the calculated current densities are intra-band current densities on graphene. Presented calculation results do not seem to support the main claim of the manuscript although the intra-band current density maps on graphene look pretty.*

The calculated currents are, indeed, arising from the intra-band dynamics of electrons in graphene. However, in pure symmetric graphene one can only observe instantaneous current but not tunneling current which we believe is measured in our experiment. To include the possibility for a tunneling current to occur in our system, we use the CAP which mimics the Si-Gr interface (see the last paragraph of the Methods section). Accordingly, we do, although phenomenologically, simulate the Si-Gr interface and observe the tunneling current in our simulations (see Extended Data 2b). Without this interface, the intra-band current in graphene is found to be zero as expected in simulation shown in Extended Data 3b.

4- *If the current is a field-driven current, I would expect it has strong directionality with respect to the polarization direction of the laser. Have they observed anything like that?*

Yes, we observed that measured current I_E current is sensitive to the polarization direction change. When we changed the polarization we observed a significantly reduced I_E ; however, it did not vanish entirely because our pump doesn't have perfect pure linear polarization (our measurements indicate a polarization ratio of 85% to 15%), due to the nonlinear propagation nature for generating the pulse in hollow-core-fiber. Moreover, we introduced a polarizer to improve the beam's linear polarization. However, the polarizer (10 mm thick) introduced temporal broadening of the pulse to approximately 200 femtoseconds, rendering the results inconclusive. This is because the observed signal disappearance could either result from the extended pulse duration or the change in polarization. As an alternative, we employed a thin quarter-wave plate to generate a circularly polarized beam. In this case, the current oscillation I_E signal disappeared then reappeared again when switching back to a linearly polarized beam.

5- Also, the photon energy is above the bandgap of silicon, so I am not sure how the photo-excited carriers are not dominating. If the photon-energy of the light is smaller than the indirect bandgap of silicon, then it would be easier to understand.

The reviewer is correct, the photo-excited carriers dynamics in graphene (interband current I_p) has the main contribution in the measured light-induced current signal. In addition, we would like to emphasize that the measured I_E current in Fig. 2 is done after cancelling out the I_p major contribution by applying external voltage with opposite sign as explained in the revised version of our manuscript in page 5.

⇒ “Accordingly, we modified our setup by splitting the input laser beam into two beams using a beamsplitter (Fig. 2a); each beam power has been set to have a similar estimated low field strength of ~ 0.85 V/nm. Then, we recorded the current as a function of the time delay between the two pulses. The recorded current, when the two pulses are not overlapping in time, is 75 nA. Our setup's capability enabled the compensation (cancelling out) of the I_p current signal (generated by the interband dynamics)—which has the main contribution in I_L —by applying an external voltage (V_{ext}) value until the output measured current is zero amperes.”

Also we identified that the I_p has the main contribution in different position in the text of the revised manuscript

i.e. page 5

⇒ “Our setup's capability enabled the compensation of the I_p current (generated by the interband dynamics)—which has the main contribution in I_L —by applying an external voltage (V_{ext}) value until the output measured current is zero amperes.”

page 6

⇒ “The photo-induced I_p current signal has the main contribution to the I_L current.”

page 7

⇒ “Next, we studied the effect of the light-induced current and carrier excitation on the resistivity and photoconductivity, mainly driven by I_p , of our phototransistor⁵⁹.”

In overall, in the present manuscript, while I find interesting experimental observations, I feel that the manuscript lack a clear interpretation of the data, and I do not find any similarities with IV curves of a typical transistor.

We hope the reviewer finds the revised interpretation and updated figures in the manuscript satisfactory. We sincerely appreciate the reviewer's insightful comments, which have greatly contributed to improving the clarity and overall presentation of both the text and figures.

Reviewer #3 (Remarks to the Author):

Key results

A) The photoresponse of the planar-graphene tunneling phototransistor designed in this study is a result from the tunneling of photoexcited carriers between the graphene layers of a Gr-Si-Gr transistor.

B) Two components contributing to the total photoresponse are the direct tunneling of photoexcited charge carriers and the field-induced tunneling current which is carrier motion driven directly by the light field itself. The latter is attosecond scale response.

C) There is a notable photoconductivity enhancement at higher light field strength

D) This phototransistor is at the technology readiness level for developing attosecond and lightwave quantum optoelectronics.

Comments on A)

Based on the current profile in Fig 1C and also the geometry of the device being graphene separated by a gap, it is reasonable to believe the current is due to tunneling photoexcited carriers. This device response is also consistent with other non-photoexcited lateral graphene tunneling devices (J. H. Yang et al., "Geometrically Enhanced Graphene Tunneling Diode with Lateral Nano-Scale Gap," IEEE Electron Device Letters, vol. 40, no. 11, pp. 1840-1843, Nov. 2019, doi: 10.1109/LED.2019.2940818.).

We thank the reviewer for their valuable comment and for bringing the mentioned reference to our attention, which supports our findings. This important work (Ref. 57) has been cited in the revised version of the manuscript.

However, clarification on device configuration would be helpful. It would be helpful to the reader if the schematic in Fig S1a reflects the actual geometry of the device with the tunneling junction in the middle along with its dimensions. The classification of this being a graphene-silicon-graphene photoresistor is also unclear, since the methods described in the text for creating the devices involve removing the middle section of the graphene and SiO₂ which means the Si is not in contact with the graphene in the lateral direction. Based on this description it would make the device a Graphene-vacuum/air-Graphene device. This may or may not affect the simulation outcomes. It would also be helpful to the reader to have the focused laser spot size and location described along with the tunneling junction dimensions. The size of the laser spot and also its location is important in understanding the origin of the tunneling current. For example, if the laser is focused such that it is only on one side of the graphene junction, then one might expect to only have field-induced current in one direction, so either it is always positive or always negative. This would help with visualizing the behavior of the field induced current.

We sincerely thank the reviewer for their valuable comments and suggestions.

In response, Fig. S1a and its caption have been revised as recommended. The updated illustration now clearly indicates the junction size, as well as the location and diameter of the laser beam. The silicon junction is formed using a tightly focused high-power laser beam (via a short focal length focusing mirror), which depletes the SiO₂ layer. Consequently, the graphene is likely in direct contact with the silicon junction at the edges as shown by Raman spectroscopy measurements.

Additionally, the laser beam illumination on both sides of the graphene is asymmetric, as evidenced by the results in Fig. 1c (where the IV curve is not symmetric around zero voltage)

and Fig. 2b (where the oscillations in the positive direction are larger than those in the negative direction). We strongly believe this asymmetry is the key factor enabling the tunneling current to flow, allowing us to measure it.

This now clarified in page 4 in the revised manuscript

⇒ Moreover, the asymmetric of this obtain IV curve (around $V_{ext}=0$ in Fig. 1c) suggests that the illumination of the laser beam of the two graphene sides is uneven (see Fig. S1a), which cause the symmetry breaking and explains the flow of the light-induced current in our setup.

Moreover, the field-induced current flows in both directions of the graphene, as demonstrated by the results in Fig. 2a (now Fig. 2b in the revised version), where the current modulation is observed in both directions (see also the illustration below).

Comments on B)

The tunneling configuration describe in this manuscript is expected to capture the field-induced current generated in graphene, since the junction would capture any charges moving into the junction and not away from it thus breaking the symmetry of the field-induced motion of the carriers. Moreover, this field-induced current has been observed previously in graphene under similar field intensities as such one would expect a field-induced current to contribute to the total photoresponse in the configuration as well. A petahertz responsive device with a simple easily reproducible geometry is indeed instrumental in studying sub-femtosecond dynamics.

On the other hand, more clarification maybe needed when describing the origins of the oscillating behavior of the field-induced current. The method in which the field-induced current was extracted was by removing the photo-induced current. This was done so by applying an external voltage across the contact of the device such that the total current is zero when the laser pulses are not overlapping. However, the magnitude of the photo-induce current could potentially vary when the laser pulses overlap due to the interference of the pulses which mean the photo-induced current may not be completely removed by applying a fixed voltage across the device. This would lead to a photo-induced current which also varies in time-delay of the pulses as the average number of excited carriers changes when the pulse interfere constructively or destructively. Thus, this current would also modulate similarly to the cross-correlation profile of the pulses. An explanation of the potential changes in the photo-induce current, or lack thereof, would be helpful to further solidify the conclusions of the field induced current.

We are grateful for the reviewer's very valuable comments and suggestions. The reviewer expectation is correct there are minimal contribution from the optical interference and the change of the IP value on the current signal oscillation shown in Fig. 2. To quantitatively evaluate this interference effect, we measured the change power of the two beams as a function of the delay

between both of them. The results shows that a change of $\pm 10\%$ power oscillation. Since our graphene dynamics remains in the linear regime (single photon excitation) as shown in Fig. 3, the current modulation due to optical interference should be in the same level ($\pm 10\%$). The minimal effect from the optical interference is because the two beams are non-collinearly propagated and has small incident angle ($< 5^\circ$) on the graphene transistor.

In response, we have considered the reviewer's suggestions and revised our explanation of the induced field current (I_E) results in the revised version on Pages 4 and 5. These revisions provide a clearer explanation of the I_E and discuss the effect of the I_P change on the modulation of I_E and the switching behavior, as follows:

⇒ **“The current tunnelling flow mechanism gates the generated current signal in time and allows us to measure and distinguish the instantaneous field-induced current (I_E), which is generated due to the intraband dynamics in graphene. This current evolves during the laser pulse's existence time window. Hence, to measure I_E in real-time, we opted to perform a cross-correlation current measurement between two current signals generated by two pump laser pulses. Accordingly, we modified our setup by splitting the input laser beam into two beams using a beamsplitter (Fig. 2a); each beam power has been set to have a similar estimated low field strength of ~ 0.85 V/nm. Then, we recorded the current as a function of the time delay between the two pulses. **The recorded current, when the two pulses are not overlapping in time, is 75 nA.** Our setup's capability enabled the compensation of the I_P current (generated by the interband dynamics)—which has the main contribution in I_L —by applying an external voltage (V_{ext}) value until the output measured current is zero amperes. The average of three first-order cross-correlation current measurements is shown in Fig. 2b (black dots connected with red lines). The absolute measured I_E current amplitude signal in real-time (plotted in Fig. 2c) switches from 29 nA (ON status) to < 1 nA (OFF status) in 630 attoseconds (see the inset of Fig. 2c), demonstrating the attosecond current switching in our phototransistor. **Please note, the modulation of the I_E oscillates in both negative and positive sides indicating that the I_E flows from the two graphene sides to the silicon junction every half-cycle of the driver field (depending on the driver field direction) causing the switching in subfemtosecond time window.****

On another note, a minor contribution of the current amplitude oscillation (Fig. 2b) potentially originated from the optical interference, since we observed only a 10% oscillation in the power between the two pulses at the temporal overlap. Please note the two beams aren't collinearly propagating, and they incident on the sample with small angles ($< 5^\circ$) (Fig. 2a), which minimize the optical interference effect. **Hence, we can estimate that 10% of the current modulation (shown in Fig. 2b) may come from the optical interface and the I_P change. Although, the contrast of the petahertz switching would remain in the range of 1 to 25 nA.**”

Comments on C)

Increase of the photoconductivity with intensity is expected due to excitation of carriers into the conduction band, this is also confirmed with the model calculations which seems reasonable. This gives a good metric for determining the efficiency of the logic circuit since it is expected to operate at high field intensities.

We would like to thank the reviewer for illustrating our results and related applications.

Comments on D)

The technological readiness of this device would be clearer when the comments for the extraction of the field-induced current is addressed. The fabrication of the device also appears to be able to be scaled up given that a high-power scanning laser could produce the tunneling junction required. It would be helpful to give examples of methods to distinguish the ultrafast-intra-band current response from the slower overall inter-band current. As in how would one suppress the I_E signal which changes sign to achieve an ultrafast logic gate states as described in table 1 and 2.

We would like to thank the reviewer for the helpful suggestion, which we have incorporated into our revised manuscript on page 7. The revised text now reads:

⇒ “Furthermore, the petahertz logic gate can be demonstrated by using I_E . In this case, the delay between two laser pulses τ is the input signal and the total I_E is the output signal (see Fig. 2b). When the delay between the two pulses $\tau= 360$ as, the output I_E signal=0. Alternatively, when the delay between the two pulses $\tau= 0$ as, the output I_E has 29 nA (see Fig. 2 a) which present the status 1, demonstrating the NOT gate as summarized in table 4, open the door for establishing ultrafast optical computers.”

and we provided and the new suggested example in table 4

Table 4

Signal A: Delay τ	Signal B: I_E
OFF 0	ON 1
ON 1	OFF 1

Table 4. Demonstration of the NOT logic gate. I_E is ON when the two pulse induced laser pulses are perfectly overlapped ($\tau = 0$). If the pulses are delayed by 630 as the output I_E is zero.

We sincerely thank the reviewer for their thoughtful comments and valuable suggestions, which have been instrumental in improving our manuscript. We believe the revisions have greatly enhanced the clarity and presentation of our work. We hope the revised version meets the reviewer’s expectations, and we kindly request their recommendation for publication.

Response to reviewers' comments – NCOMMS-24-50952A

The referees' comments are shown in bold and *Italics*, followed by the corresponding response in blue and the change in the revised version.

REVIEWER COMMENTS

Reviewer #1 (Remarks to the Author):

The authors adequately replied to many of my comments. I now understand physics better. The directional current in Fig. 1b is produced due to the unequal laser intensities on the two graphene-silicon interfaces. The offset of 75 nA in Fig. 2b is canceled by an external DC voltage, and therefore the direction of the measured current (29 nA amplitude) could be reversed. On the other hand, there are two remaining points which should be made clearer before publication.

1. The structure of the sample is still not clear to me. The authors removed graphene and SiO₂ and exposed the silicon substrate. How can it make a graphene-silicon interface? From a three-dimensional perspective, isn't there at least 90 nm distance (=SiO₂ thickness) between graphene and silicon?

The formation of the Si junction is achieved by focusing an intense laser beam on the center of the chip to deplete the graphene and SiO₂ layers. The chip, mounted on a translation stage, is then moved along the x-axis. The laser beam follows a Gaussian intensity distribution, with the highest intensity at its peak, gradually decreasing toward the edges. As a result, depletion occurs in an arc shape, corresponding to the most intense region of the beam. Additionally, we anticipate that the Si melts and fills this arc due to the high laser power. Consequently, the graphene and SiO₂ layers may be connected, as confirmed by our Raman spectroscopy. The illustration below depicts the formation of the junction.

We appreciate the reviewer's comment! In response, we have clarified this point in the revised version of the SI and included this illustration in Fig. S1.

2. Regarding the ‘first-order’ cross-correlation, I mentioned ‘cross-correlation’ at a different place, which is in the sentence ‘In our calculation, ... cross-correlation of the laser fields’. The question was asked because it was a bit of a surprise to me that the observed result in Fig. 2b can well be approximated by the first-order cross-correlation of the laser fields, meaning that the switching speed of the device is solely determined by the cycle period time of the laser field.

The authors excluded the contribution of optical interference (reply to reviewer #3) but the revised manuscript states that 7.5 nA (= 10% of the 75 nA) of the 29 nA can come from the optical interference. On this point, the authors need to prove more carefully that the dominant effect is still the tunning, not the optical interference.

My specific comments are

(2-a) An explanation should be added as to how and where the 10% power modulation was measured. Since two non-collinear beams can produce delay-dependent interference fringes in space, the local intensities on the graphene-silicon interfaces can be modulated by more than 10%. Since the current is driven by unequal local intensities, the interference fringes may cause a strong influence. Did the authors exclude this possibility, too?

We would like to thank the reviewer for their constructive comment!

Regarding the power modulation measurement, it was conducted in the same plane as the graphene transistor, where the measurements presented in Fig. 2a were taken. In response to the reviewer's comment, we have included this information in the revised version of the manuscript on page 5.

On another note, we replaced the phototransistor by a power meter, and we observed only a 10% oscillation in the power between the two pulses at the temporal overlap, indicating a minor contribution of the current amplitude oscillation (Fig. 2b) potentially originated from the optical interference.

Regarding spatial interference, its effect is minimal due to the very small angle ($<5^\circ$) between the two beams. To verify this, we measured the beam profile using a beam profiler at the temporal overlap (at the focus plane, where the experiment takes place). The snapshots show no significant interference fringes in space (red circle).

Please note that the beam structure highlighted by the white square results from using a parabolic mirror to focus the beam. The astigmatism effect occurs if the beam is not perfectly perpendicular to the mirror. However, this does not affect the current measurement since the focused beam in the red circle almost entirely covers the area of the graphene transistor.

Additionally, to demonstrate the minor contribution from optical interference, we conducted the experiment using a chirped laser pulse. The results are now presented in Extended Data Fig. 1 of the revised manuscript. Here, the current modulation is asymmetric at positive and negative delay sides, indicating that optical interference has only a minor contribution. If optical interference were the dominant effect, one would expect perfectly symmetric modulations on both sides (negative and positive).

Extended Data Fig. 1. The measured cross-correlation I_E current modulation driven by a field of chirped few-cycle pulse.

(2-b) Is the noncollinear geometry considered in the theory with the dipole approximation? If not, the theory gives a strong optical interference effect and cannot support the interpretation. Additional explanation is needed on this point.

In our theory simulations, we used a simple theoretical model that can only take the amplitude and the polarization of the applied laser fields into account. Therefore, we cannot simulate scenarios like the noncollinear geometry requested by the reviewer. Furthermore, we believe these additional layers of complexity will not provide anything new for understating the nature of the observed phenomena. Taking more effects into consideration will only complicate the Hamiltonian of the system. However, all the observed physics comes from the analysis of the wavefunction/density matrix which still will contain information about both the optical interference and the generated tunneling current, as well as their possible dependence on each other.

Yet, there is a simple way we can distinguish the electric current generated due to the optical interference from that arising from the tunneling effect. We thank the reviewer for their stimulating comment which inspired us to perform additional calculations discussed in the revised main text page 6 and the last paragraph of the Methods section which we copy below.

“The persistent electric current obtained via solution of Eqs. (1) and (7) with the field synthesized from the two delayed Gaussian waveforms can, in principle, be attributed to two concurrent effects: optical interference of the applied fields and electron tunnelling through Gr-Si junction. To decipher these two mechanisms from each other, we performed additional simulations selectively switching off appropriate terms in Eq. (1) and analysing the obtained current. Therefore, in addition to two terminal cases described above, (a): full system with possibility of tunnelling and optical interference (Extended Data Fig. 3a), and (d): pure graphene with no junction (Extended Data Fig. 3d), we created two intermediate scenarios where we (b) prevent photon absorption by neglecting $\mathbf{E}(t) \cdot \{\mathbf{D}(\mathbf{k}_t), \rho(\mathbf{k}, t)\}_{m,n}$ term. This way we exclude any optical interference effects since the obtained density matrix will not, by definition, contain any contributions coming from the interaction with an optical field (Extended Data Fig. 3b).

Alternatively, we (c) prevent field acceleration of electrons by prohibiting changes of the crystal momentum frame in time: $\mathbf{k}_t = \mathbf{k}_0$. This way we exclude tunnelling effects since the electrons will not be able to move through the Gr-Si junction (Extended Data Fig. 3c). As one can see, exclusion of the optical excitations and thus the optical interference from the consideration (Extended Data Fig. 3b) has a minor effect on the possibility to generate the electric current. This is because the number of electrons driven to the conduction band by the photon absorption is negligible in comparison to those already present there due to the thermal effects (see Eq. (3)) or originating from the Landau-Zener transitions. At the same time, excluding acceleration of electrons by the field and thus dramatically reducing the possibility of tunnelling, we see nearly complete suppression of the generated current (Extended Data Fig. 3c)”

We would like to thank the reviewer once again for their time and valuable comments, which have helped us improve our presentation and simulation. We sincerely appreciate the reviewer’s recommendation to publish our manuscript in Nature Communications

Reviewer #3 (Remarks to the Author):

The follow up comments below should be addressed before publication can be recommended.

The main claims of the manuscript is restated below for reference, follow up comments follows.

A) The photoresponse of the planar-graphene tunneling phototransistor designed in this study is a result from the tunneling of photoexcited carriers between the graphene layers of a Gr-Si-Gr transistor.

B) Two components contributing to the total photoresponse are the direct tunneling of photoexcited charge carriers and the field-induced tunneling current which is carrier motion driven directly by the light field itself. The latter is attosecond scale response.

C) There is a notable photoconductivity enhancement at higher light field strength

D) This phototransistor is at the technology readiness level for developing attosecond and lightwave quantum optoelectronics.

Follow up Comments on A)

The added scale for device in the revised manuscript is helpful. However, the overall geometry of the device is still unclear. I am unsure as to why the Graphene would be in contact with the silicon since the graphene should be on top of the SiO₂ from the fabrication process. A cross-section schematic of the structure would be helpful here (see attached example). As for the Raman spectroscopy, while the silicon and graphene peaks are shown, that could potentially mean both the graphene and silicon are within the beam spot of the laser, but they could be vertically displaced.

The formation of the SiO₂ junction is achieved by focusing an intense laser beam on the center of the chip to deplete the graphene and SiO₂ layers. The chip, mounted on a translation stage, is then moved along the x-axis. The laser beam follows a Gaussian intensity distribution, with maximum intensity at its peak, gradually decreasing toward the edges. As a result, depletion occurs in an arc shape, corresponding to the most intense region of the beam.

Additionally, we anticipate that the Si melts and fills this arc due to the high laser power. Consequently, the graphene and SiO₂ layers may be connected, as confirmed by our Raman spectroscopy scan. The illustration below depicts the formation of the junction.

We appreciate the reviewer's comment! In response, we have clarified this point in the revised version of the SI and included the cross-sectional illustration in Fig. S1.

“The SiO₂ junction is formed by a laser-induced depletion process. A focused laser beam with a Gaussian intensity profile is incident on the center of the chip, targeting the multilayer structure of graphene and SiO₂. As the chip is translated along the x-axis, the central, high-intensity portion of the laser beam interacts with the material, causing localized depletion. This depletion results in a arc-shaped region where the graphene and SiO₂ layers are effectively removed (see cross-section illustration in Fig. S1). The formation of this junction is further supported by Raman spectroscopy scans, as we explain in the next section”.

Follow up Comments on B)

Since the light is exciting both graphene and silicon, is there a difference between the carriers tunneling from silicon to graphene vs graphene tunneling to silicon?

The tunneling direction depends on the direction of the excited field. As a result, electrons first tunnel from one side of the graphene to the silicon and then from the silicon to the opposite side of the graphene. This explains the observed current modulation in both positive and negative directions, as shown in Fig. 2b—otherwise, the current would be unidirectional. The measurements presented in this work suggest no significant difference between Si/Gr and Gr/Si tunneling.

It is also worth noting that the valence and conduction bands in graphene meet at the Dirac point, whereas silicon has a band gap of 1.12 eV. Consequently, the number of electrons in graphene's conduction band—both thermal electrons and those generated by Landau-Zener transitions—is significantly higher than the number of electrons that could be excited in silicon.

Also, in reference to the polarization testing mentioned by reviewer #1. It is not immediately clear why the circular polarization does not produce the same variation with since you can consider it as two orthogonal linear polarizations (one parallel to the stripe, and another perpendicular to the stripe).

This is because the test was conducted by rotating the quarter-wave plate while keeping the input beam power constant. As a result, the power is distributed between the two linear polarization components, leading to a lower induced-field current signal.

Lastly the linear polarization dependence is strong experimental evidence for this effect being field driven and should be included in the main text.

We appreciate the reviewer's suggestion! We have revised the manuscript and added the following statement on page 5:

“In addition, the induced effect (I_E) is polarization-dependent, as the signal diminishes when the input linearly polarized beam is converted to circularly polarized light.”

Additionally, we have included Extended Data Fig. 1, which shows asymmetric current modulation induced by a chirped pulse, as further evidence for the field-driven current measurement.

Extended Data Fig. 1. The measured cross-correlation I_E current modulation driven by a field of chirped few-cycle pulse.

Finally, we would like to thank the reviewer once again for their valuable comments during both the first and second rounds. We hope we have adequately addressed all the raised questions and comments. We appreciate the reviewer's recommendation to accept our manuscript for publication in *Nature Communications*.

Response to reviewers' comments – NCOMMS-24-50952B

The referees' comments are shown in bold and Italics, followed by the corresponding response in blue and the change in the revised version.

REVIEWER COMMENTS

Reviewer #1 (Remarks to the Author):

Reviewer #1 (Remarks to the Author):

The authors have adequately responded to my last comments. But the new data, Extended Data 1, is difficult to understand. If my understanding is correct, the two pulses are nearly identical. If so, the same signals must be observed at plus and minus delay times because the two pulses are indistinguishable regardless of the chirp and the nonlinearity involved. If the two pulses are significantly unequal, the new result would not strongly support the claim. Additional explanation(s) should be added.

The measurement shown in Extended data Fig. 1 was performed using a 6.5 fs pulse (pulse 1) in one beam and a chirped pulse (pulse 2) in the other beam. We deconvoluted the measured trace in Extended Fig. 1 and plotted the results in the following figure: in (a) the waveform of the 6.5 fs pulse (pulse 1) and in (b) the chirped pulse (pulse 2). The calculated cross-correlation of pulse 1 & 2 (red line in (c)) is in acceptable agreement with the measured current modulation in Extended Data Fig. 1.

Fig. R1: The retrieved waveforms of the two pulses used to obtain the cross-correlation current modulation in Extended data Fig. 1: (a) pulse 1 (6.5 fs) (b) pulse 2 (chirped pulse). (c) The calculated cross-correlation of the two pulse is plotted in red in comparison with the measured current modulation presented in the Extended Data Fig. 1.

We revised the manuscript accordingly to clarify this point in page 3

“Furthermore, we measured the I_E cross-correlation using a chirped input pulse in one beam and the 6.5 fs pulse in the other beam”

Reviewer #3 (Remarks to the Author):

The comments on two of the main claims in the manuscripts was addressed in the previous rounds of comments. However more clarification is needed before publication to support the following two claims listed below (restated from the first round of comments for reference).

A) The photoresponse of the planar-graphene tunneling phototransistor designed in this study is a result from the tunneling of photoexcited carriers between the graphene layers of a Gr-Si-Gr transistor.

B) Two components contributing to the total photoresponse are the direct tunneling of photoexcited charge carriers and the field-induced tunneling current which is carrier motion driven directly by the light field itself. The latter is attosecond scale response.

(Follow up comments on A)

Both reviewer #1 and myself asked for clarification of the structure, which the authors has provided. While the provided explanation could explain the formation of the structure it is only a speculation on the melting of the Si. Perhaps an AFM measurement of the surface can show this effect.

We tried to perform the AFM measurement earlier however the structure of the commercial transistor (shown in the following photo) is covered by a black protection layer. The height of this layer prevented the AFM tip to reach the surface of graphene. We tried to remove the black protection layer without damaging the graphene structure, but we failed to remove it. Worth mentioning, our explanation of the structure is supported with the performed and provided Raman measurements and results.

The protection
black layer

(Follow up comments on B)

A clarification on the asymmetric time delayed data using a chirped pulse, and a demonstration of the time-delay photocurrent data with different polarization is needed to support this effect being field driven.

A more clear explanation on why an asymmetric current modulation implies a field driven effect is needed. This asymmetry suggests the charge carriers cares about which of the two identical pulses arrives first, which intuitively even with a field driven affect should not occur.

We refer the reviewer to our answer to the reviewer 1 comment on the same point, in page 1.

Data showing the diminished tunneling current with respect to excitation by a pulse with linear-polarization parallel or semi parallel to the channel as mentioned in a previous rounds of comments should be included in the manuscript for this would be one of the hallmarks of this effect being field driven.

We included the requested data in the revised version in Fig. S3.

Fig. S3: The measured modulated current using circular polarized pulse.

Response to reviewers' comments – NCOMMS-24-50952C

The referees' comments are shown in bold and Italics, followed by the corresponding response in blue and the change in the revised version.

REVIEWER COMMENTS

Reviewer #1 (Remarks to the Author).

- ◆ **Regarding the new information that a combination of chirped and un-chirped pulses was used for Extended Data 1, I cannot follow why and how the asymmetry in the data is connected to their conclusion of “minor effect from the optical interference”. Optical interference can give asymmetric data because one of the pulses is asymmetric in time. A clear explanation should be added before publication.**

⇒ We agree with the reviewer that the asymmetry observed in the measurement shown in Extended Data Fig. 1 of the previous version (now Supplementary Fig. 4) may not fully rule out the possibility of an interference effect, which we acknowledged in our discussion in the main text. However, the results suggest that the measured current is field-sensitive. In response to the reviewers' suggestions, we have added the requested explanation to the revised manuscript (page 5).

“Furthermore, we measured the I_E cross-correlation using a chirped input pulse in one beam and the 6.5 fs pulse in the other beam and the results are shown in Supplementary Figure 4 indicating that the I_E is field sensitive.”

- ◆ **In addition, the authors should describe in the method section how the pulse was chirped.**

⇒ We have considered the reviewer's suggestion, and the following sentence has been added to the Methods section page 9:

“For the measurements shown in Supplementary Figure 4, one of the pulses is chirped by propagating it through a thick piece of dispersive fused silica”.

- ◆ **If the asymmetry in data cannot exclude the possibility of optical interference, the measurement result with a power meter (10% modulation) is the only direct evidence. To strengthen the authors' claim, I suggest adding a description of how this number (10%) is translated into the amount of modulation of I_E . If the modulation of I_E is proportional to the laser intensity, I_E should then be modulated by 10% due to optical interference. However, if I_E varies nonlinearly with laser intensity, optical interference causes stronger modulation.**

⇒ As shown in Fig. 3, the measurements were conducted in the linear regime, where power (P) is directly proportional to intensity (I), according to the relation $I=P/ A$, where A is the area of the beam. Consequently, a 10% change in power results in a corresponding 10% change in intensity.

In response to the reviewer request we added the following in the revised version of the manuscript (page 5):

“Moreover, the measurements were conducted at low power in the linear regime, where power is directly proportional to intensity. Hence, we can estimate that 10% of the current modulation (shown in Fig. 2b) may come from the optical interface and the I_P change.”

◆ **Lastly, although the authors replied to referee #3 and me previously that the connection between graphene and Si was confirmed by Raman measurements, I cannot find any proof that the SiO₂ layer was removed. There is no Raman peaks assigned to SiO₂ in Fig. S2. The authors should add an explanation before publication. If the removal of the SiO₂ layer is a hypothetical level, it should be expressed as such.**

⇒ We considered the reviewer’s suggestion and revised the corresponding section (Note 2) in the supplementary information to clarify the Raman measurement results. The updated version can now be found on page 8 of the SI:

“The graphene peaks are entirely absent in the spectrum of the junction area in the modified device, indicating a clear displacement of the graphene layer and likely suggesting the removal of the SiO₂ layer, thereby exposing the silicon substrate.”